# A Study on the Design and Control of the Overhead Hoist Railway-Based Transportation System

Thuy Duy Truong [1,2,3], Xuan Tuan Nguyen [1,2,3], Tuan Anh Vu [1,2,3], Nguyen Huu Loc Khuu [1,2,3], Quoc Dien Le [1,2,3], Tran Thanh Cong Vu [2,3], Hoa Binh Tran [2,3] and Tuong Quan Vo [1,2,3,*]

[1]  Department of Mechatronics, Faculty of Mechanical Engineering, Ho Chi Minh City University of Technology (HCMUT), 268 Ly Thuong Kiet Street, District 10, Ho Chi Minh City, Vietnam
[2]  Bach Khoa Research Center for Manufacturing Engineering, Ho Chi Minh City University of Technology (HCMUT), 268 Ly Thuong Kiet Street, District 10, Ho Chi Minh City, Vietnam
[3]  Vietnam National University Ho Chi Minh City, Linh Trung Ward, Thu Duc City, Ho Chi Minh City, Vietnam
*  Correspondence: vtquan@hcmut.edu.vn; Tel.: +84-933-32-7078

**Abstract:** Overhead hoist transportation systems (OHTS) have been the subject of worldwide research and development in recent years. The majority of these systems are utilized in semi-automated or fully automated factories. This article proposes a new solution for OTHS based on the concept of the modulation of mobile units that can move on a railway structure from one point to another. The OHTS mentioned in this article is a group of shuttles that can operate independently but which also have the ability to cooperate together to complete the desired tasks. By using the space below the ceiling, this system can operate without affecting the original design of the factories. There are many potential fields of application for picking-up and delivering, such as the medical field, the food and beverage fields, automotive and electrical appliances, etc. Moreover, by applying Dijkstra's algorithm in the controller design, the transportation speed among the stations in the whole system can be improved. The real prototype of the whole system, including three shuttles, is also manufactured to explore and assess the design and operation of the proposed system and its controller.

**Keywords:** hoist; rail system; shuttles; task; control; station; elevator

## 1. Introduction

According to Daniel Küpper et al., the Boston Consulting Group (BCG) proposed the installation of smart robots and the use of collaborative robots in future factories [1]. This article focuses on the introduction of overhead hoist transportation systems (OHTS) and the new methodological aspects of mechanical features, position detection, and algorithms in the controller design. The proposed system is being utilized in a variety of fields— i.e., general manufacturing facilities, warehouse operations, or medical equipment in hospitals. It has the ability to aid employees and enhance the production procedure. Nowadays, the logistics industry is experiencing increasingly intense competition due to the development of global e-commerce. The advantages of providing logistics services more quickly will lead to a greater market share. The application of an automated guided vehicle (AGV) increases the productivity of the last workstation of an assembly line by implementing an AGV system to transport finished goods to the warehouse according to Nunno Correia et al. (2020) [2], or the research of Andrea Ferrara et al. on the pallet shuttles in automated warehouses [3]. In a similar fashion, warehouse conveyor systems are one of the solutions that can help a warehouse run on time and minimize accidents according to Hector Sunol of the Cyzerg company [4]. However, these solutions mostly develop the transport network on the ground. As a result, the search for multi-station multi-shuttle transportation systems (MSMSTS) represents an outstanding innovation in the logistics sector. Especially in modern-day Vietnam, investment capital is still low,

Vietnam's mechanical industry is still quite backward, labor resources and raw materials are not of good quality, and their operation is still quite weak [5].

Hazardous materials are crucial for harvesting industrial raw materials and economic development in many countries. Nevertheless, the use of these technologies can lead to potential hazards for human beings, assets, and natural surroundings. The successful management of hazardous materials (hazmat) logistics necessitates the seamless integration of various factors, including transportation expenses, safety considerations, and the strategic placement of warehouses together with efficient inventory control. Hazardous materials can be classified into three main categories: flammable, toxic, and corrosive compounds. The logistics costs and associated risks vary among these classifications. This study examines the design of distribution networks for hazardous materials, with a focus on the integration of several classes of hazardous materials and the analysis of the interaction between the different types. The consideration of time-dependent road closures and detours is of utmost importance in the design of hazardous distribution networks, as it entails substantial financial commitment and has enduring consequences. In their study, Wu, Weitiao, and colleagues presented a problem called the multi-class hazmat distribution network design problem (MHND) in a multi-echelon supply chain. This problem takes into account the presence of overlapping risks associated with several types of hazardous materials [6]. The model's scalability allows it to be used across many types of hazardous materials (hazmat), which has significant consequences in the field of hazmat logistics. This paper examines a real-world scenario in Guangzhou, China, and demonstrates that substantial reductions in population exposure can be achieved while simultaneously reducing overall costs. The findings indicate that it is advisable for agencies to carefully consider and balance the factors of economic viability, exposure risk, and customer service when making decisions. The distribution network design is notably influenced by the interactions among various types of hazardous materials (hazmat), whereby the degree of interaction intensification leads to a decrease in the overlapping ratio of routes. The study additionally presents novel findings pertaining to the association between transportation risk and inventory risk, the concept of risk equilibration, the influence of service level, and the interplay among various categories of hazardous materials. Transportation expenses include both permanent and variable expenses, as well as nonrecurring expenses such as delivery management and driver scheduling. The comprehensive model emission model (CMEM) is used to calculate fuel consumption, which is connected to driving speed, driving range, and vehicle load.

The design of distribution networks poses significant challenges in areas such as location-allocation, vehicle routing, and inventory control. Comprehensive models have been developed by researchers to address these issues, including location-routing, inventory-routing, and location-routing-inventory problems. The objective of these models is to simultaneously optimize the plans for placement, routing, and inventory in a logistics system that consists of multiple echelons. The inventory routing problem (IRP) is a logistic optimization challenge that involves both inventory management and vehicle routing difficulties. The location inventory routing problem (LIRP) is an alternative to the inventory routing problem (IRP) that aims to enhance the integration of location selection, vehicle routing, and inventory control. Nevertheless, the solution space of this problem is more extensive compared to those of the vehicle routing problem (VRP) and inventory routing problem (IRP) due to its significant non-linearity and NP-hardness. Genetic algorithms, variable neighborhood search algorithms, and Tabu search algorithms are frequently employed in the resolution of IRP and LIRP models. Hybrid heuristic algorithms utilize the strengths of multiple distinct algorithms to enhance their overall performance. In the field of supply chain management, managing inventory is a crucial operational challenge. In their study, Wu, Weitiao, and colleagues introduced an improved multi-period two-echelon location inventory routing problem (LIRP) model that incorporates time windows at merchants and accounts for vehicle fuel use [7]. The proposed model simultaneously optimizes location, inventory, and routing variables, taking into account the replenishment needs of both

distribution centers and retailers over many time periods. To enhance the optimization of replenishment plans, this study suggests the use of a customized genetic algorithm and a gradient descent method. The results indicate that there are trade-offs between inventory and transportation costs. Specifically, as transportation costs and inventory holding costs decrease, there is an increase in stock-out costs. The impact of post-optimization on inventory levels in distribution centers is contingent upon the spatial distribution of retailers. It is possible to decrease or increase inventories. Additional areas for future research could involve the development of models to address uncertain and linked demand in the context of transportation, as well as exploring the application of the location inventory routing problem (LIRP) specifically for the transportation of particular items.

The world now has numerous transportation systems. First is the pneumatic tube system (PTC) [8,9]. The PTC is used to load materials of negligible load at speeds up to 5 m/s according to [8]. In [9], Shibani et al. pointed out the shortcomings of this system, such as only transferring loads of limited size with a weight of about 5 pounds. Next, Telelift GmbH is a company that has a more than 55-year-long history of experience in the development of internal transport systems [4,10], and MultiCar is a good example. MultiCar has a 3-line rail system, including two lines of 24V suppliers and one line for a signal. It uses a position code to detect the shuttle's location. The velocity of MultiCar is also significant as it can carry a mass of 60kg at a maximum speed of 0.5 m/s.

Compared to other transportation systems, Telelift's monorail system is designed to bend with the help of complex cable tray profiles. Skyrav is the newest overhead hoist transport system created by Murata Machinery USA, Inc. [11]. The overhead transport provided by Skyrav allows for the provision of different collection and deposit points; it is clearer, with a contactless power supply and more space-efficiency than other transport systems. The Murata system is made up of a double rail and does not have the modules curving along the vertical axis. In addition, the research of Martin Görner et al. supplies a new method for indoor transport and mobile manipulation by applying the Omni-wheels and continuous overhead manipulation [12]. Swarmrail Robot commutes on the rail structure, which has no transfer rails module but intersections instead. With 8 Omni-wheels, the Swarm Rail Robot changes direction at an intersection. However, all of the mentioned systems (Telelift, Skyrav, or Swarm Rail) operate on a direct power supply, and the mechanism of OHTs is still complicated and only applies to large-scale factories or warehouses.

The aim of this research is to develop a new mechanical and controller design that can be utilized in both large and small-scale factories in Vietnam. The concept of modulation is used in the mechanics and controller system, making it possible to extend the scale with more stations or shuttles. In addition, the location detection method and algorithms in the controller design are also proposed to control this system. In particular, the explanation will focus on a realistic mechanical model and a queueing algorithm. This has only been a closed-looped system in the initial phase.

The article is divided into seven sections: Section 2 introduces the basic concepts of mechanical and controller design. Details of the mechanical, electrical, and electronic design as well as the difficulties of the problem of design are provided in Section 3. Section 4 explains the algorithm used to control two major modules and the system sub-function. Section 5 shows the results of the primary simulation with respect to four different cases. Sections 6 and 7 provide descriptions of the algorithm's conclusions and a discussion about the experimental results. Section 8 exposes the final conclusions and the directions for future work.

## 2. General Concepts

### 2.1. The Mechanical Model

The proposed design of the shuttle in Figure 1 and the ray module of the system in Figure 2 propose a general description of the multi-station transportation models in the structural mechanics and system layout. This model has three mobile units which

can be used to deliver the packages among stations at the same time. In particular, it is capable of supporting more if necessary. By accessing the factory workspace from the above space, this system can enhance profits and competitiveness in the logistics and manufacturing industries.

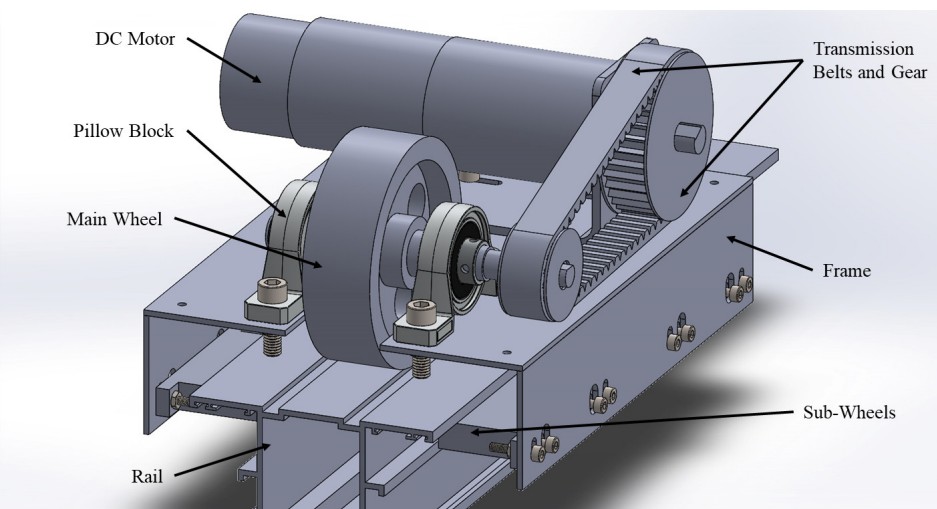

**Figure 1.** The mechanism design of the shuttle.

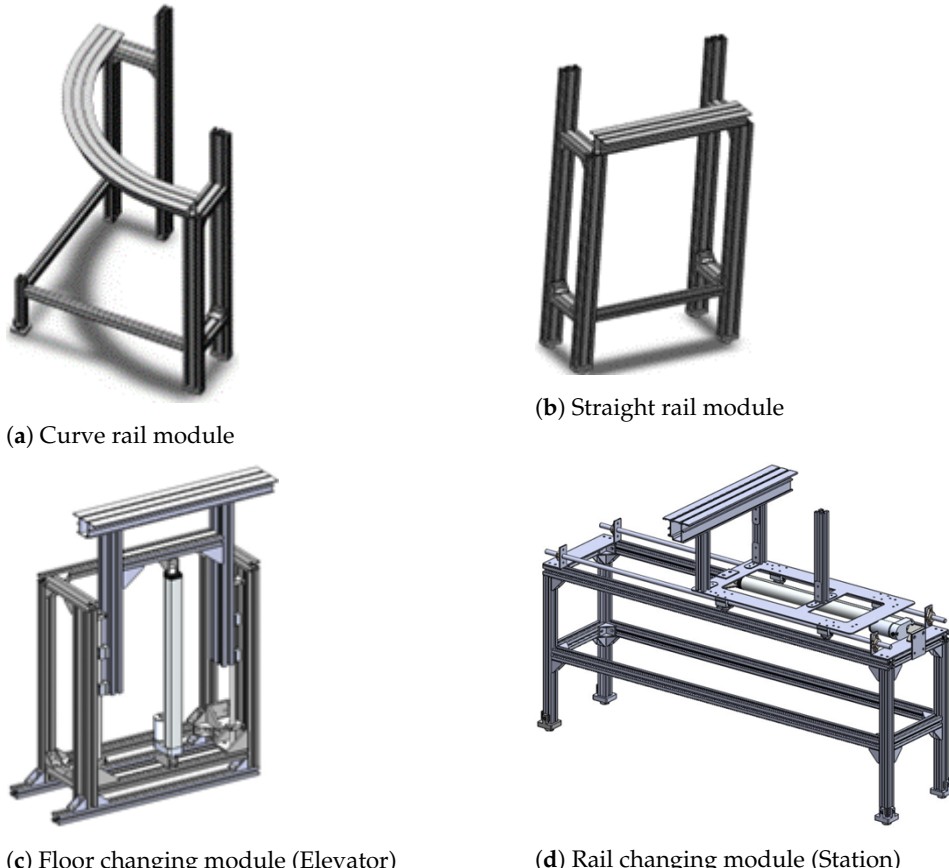

(**a**) Curve rail module

(**b**) Straight rail module

(**c**) Floor changing module (Elevator)

(**d**) Rail changing module (Station)

**Figure 2.** Design of the main elements of the rails module.

We first designed and fabricated a small model to check the modulation characteristics and controller of the entire system. The mechanical parts (rails and shuttles) are designed on a moderate scale. The rail systems' modularizations are made up of straight rails and

curved rails. The mechanism of the whole system is also designed such that its parts can work well together and can be extended to more stations and shuttles in the future.

## 2.2. The Controller Design

In order to simplify the electrical systems, the RDIF readers are employed as a slave unit to gather all signals from the RFID tags [13]. After that, the data will be transmitted to a PC to control the shuttles' operation. The RFID tag and encoder signal from the shuttles' motors are utilized to determine the relative position of the shuttles on each rail. The idea of using RFID devices was inspired by [14,15]. The authors in [14] also stated that the radio frequency of 13.56 MHz is used in the access control and payment systems, as well as for identifying goods in warehouse systems and books in library systems. The RFID system is chosen to identify containers that are moving in the system based on previous ideas [16]. The 24 V battery is chosen as the main power source for the entire shuttle.

Our goal which is shown in Figure 3, have all the shuttles carry out the required tasks and, at the same time, optimize travel distances. The route that the shuttles use for delivering can be considered a closed circle, and the shuttles can only move in two directions, which leads to the process of other shuttles being obstructed. It is therefore necessary to optimize the system in terms of time and energy.

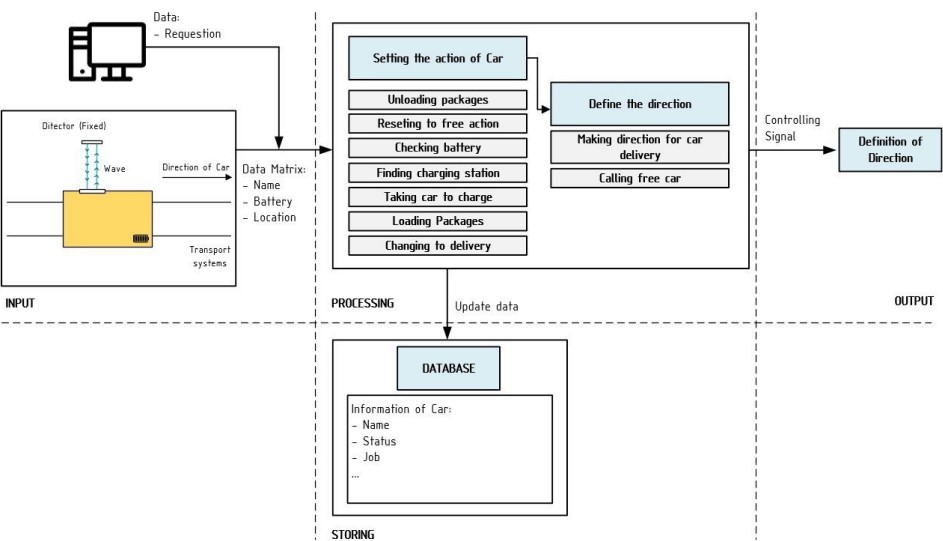

**Figure 3.** The schematic diagram of the system.

## 3. Design of the Whole System

### 3.1. Mechanical System

#### 3.1.1. The Proposed Rail Based System

To test the entire system, the rail is made of aluminum with a standard thickness of 2 mm. The rail system consists of two main modules: the straight module and the changing direction module (as introduced in Figure 2). This prototype was created and fabricated by us to verify the design concept and controller.

#### 3.1.2. The Shuttle Design

The principle diagram of the shuttle is introduced in Figure 4 below.

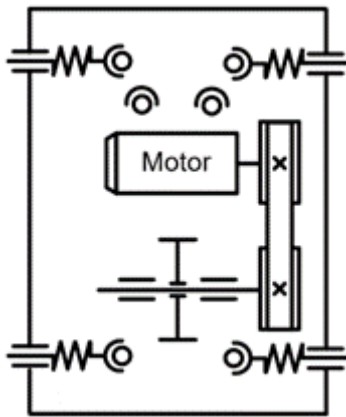

**Figure 4.** The principle diagram of the shuttle.

The drive part uses a timing belt which is affordable and easy to fabricate. Eight wheels are used to operate each shuttle, ensuring that the vehicle stays stable and does not tip over while moving on the rail. The hanger arm's ball-bearing joints ensure that the gravity vector of the goods is always in a stable state when operating. The shuttle-base design and fabrication are shown in Figure 5 below in accordance with the basic specifications introduced in Table 1.

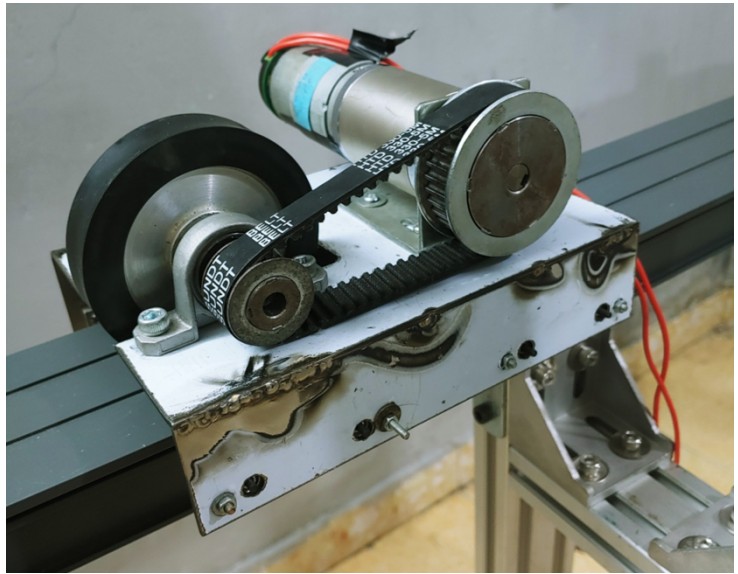

**Figure 5.** The real shuttle base for experiment.

**Table 1.** Four stages of the processing mission.

| Processing Steps | Details |
| --- | --- |
| 1 | Take the goods at load Station |
| 2 | Load the goods |
| 3 | Deliver the goods |
| 4 | Unload the goods |

The specifications of the experimental shuttle are as follows: size, 220 × 100 × 80 mm; mass, 3.5 kg; payload, 10 kg; max speed, 0.3 m/s; actuator, DC Planet Gear motor.

### 3.1.3. Elevators and Stations Design for Experimental Testing

When shuttles arrive at stations or elevators, the PC calculates and sends decision signals through the NodeMCU. The NodeMCU will immediately send commands to the

Microcontroller STM32 to activate the stations or elevators to bring the shuttle in or out of the new floor. The elevator system is utilized to move the shuttle from the current floor to the upper floor and vice versa, as explained in Figure 6 below.

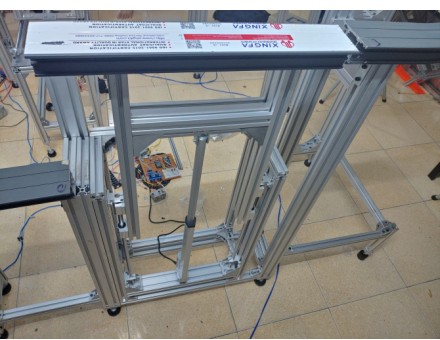
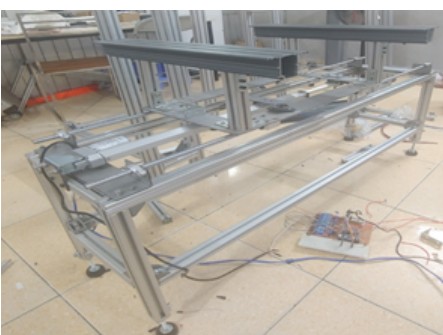

**Figure 6.** An experimental station and elevator.

### 3.2. Electrical and Electronics Design

To supply the power, a 24 V battery is used to power the driving circuit. In addition, the RFID reader is combined with a pulse counting encoder to determine the relative position of the shuttle while operating. Figure 7 is the schematic diagram of the system in use.

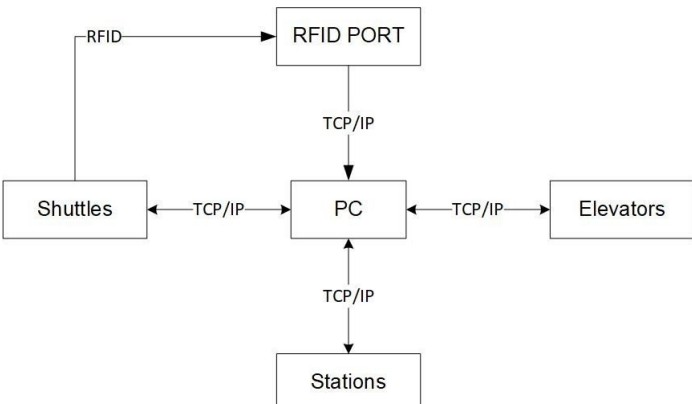

**Figure 7.** The transmission and reception diagram for the system's electrical signals.

The transmission and reception signals from shuttles, stations, and elevators to PC are described in Figure 7 using TCP/IP protocol. During their movement, the shuttles are repositioned using the additional RFID devices. In addition, encoder signals can be used to calculate the relative position of the shuttles. Therefore, it is necessary to place the shuttles' RFID tags and RFID readers on the route to reposition them. The RFID-RDM850 TCP/IP + WIFI RFID reader with a scan range of 50 mm is utilized to determine the shuttles' absolute position [17,18].

### 3.3. The Difficulties of the Problem

The railway system is viewed as a large closed circle. It is easy to cause congestion when shuttles can move in two directions (clockwise and counterclockwise). As a result, it is rather difficult to allocate shuttles when there are many trips. According to some simulations, finding the number of shuttles and stations to ensure the optimal use of energy for all cases is a difficult problem because it also depends on the location of the stations, the length of the path, etc. Building models in the real economy of factories and enterprises will make it easier to find the optimal number of shuttles and stations in terms of energy. For example, when we install this cargo transportation system in the factory, we will know in advance the specific quantity of goods that needs to be reached in 1 h. Furthermore, we

know the specific locations of freight forwarding stations (main terminals), as well as the possible locations of auxiliary stations. The number and location of stations, as well as the distance, are fixed, so we only need to determine the number of shuttles.

In terms of energy saving, it is quite difficult to find a solution specifically intended to save energy. At the same time, it may lead to the car running out of battery during travel because of too much focus being afforded to the energy saving of the entire vehicle system.

For example, Shuttle A is given the command to go clockwise at time T1. At time T2, when Shuttle A has not completed its task, an event changes the state of the system, causing the system to order Shuttle A to go counterclockwise. Then, at T3, when Shuttle A has not finished its task, a new event occurs which causes Shuttle A to operate clockwise. After a long time, Shuttle A completes its work by continuing in this manner. However, this system is designed to transport goods in accordance with the customer's requirements. For example, in a drug delivery system in a hospital, a patient is first in line to buy medicine but his medicine is delivered at the end of the day. That makes it unacceptable to customers. In handling goods, the first-in-first-out factor must be used to design the problem. This leads to an unreasonable character because it must simultaneously satisfy the energy saving factor and ensure that tasks are performed without leaving any backlog.

## 4. Control Algorithms

The use of artificial intelligence and robotics in the field of robot path planning is significant in the domain of mobile security. The task at hand entails the identification and determination of pathways that are collision-free, enabling mobile robots to successfully navigate towards their respective objectives. Path planning is divided into two distinct components: global planning and local planning. The former pertains to situations when the environment is well-known, while the latter deals with scenarios where there is limited or unknown information about the surroundings. Through simulation trials, the effectiveness of this approach has been empirically demonstrated, enabling robots to autonomously navigate maze environments. The Dijkstra algorithm proposed in this article has been shown to be accurate and efficient in mobile robot path planning. This study has found several problems, one of which is the robot's working space being a two-dimensional grid that has obstacles spanning from unobstructed areas to barriers [19]. To perceive ambient data, the robot is depicted as a single point, while barriers are depicted as linear entities. The technique uses a Cartesian spatial approach to represent the robot's movement. The robotic system is required to make decisions regarding obstacle avoidance through a series of computational steps, including line drawing, intersection detection, intersection identification, point connection, and execution of four distinct operations [19].

The line sensor location problem (SCLP), or traffic surveillance problem, is a challenging task that involves selecting a line to separate demand centers into two groups: origins and destinations [20]. The collection of census information and control of estimated origin/destination (O/D) matrices is of the utmost importance. To simplify the problem, it can be transformed into a set covering problem. The transformation is contingent on enumerating all potential paths between origin–destination pairs, which becomes infeasible in the case of vast networks. The problem of set coverage has been proven to be NP-hard. In the absence of limitations on the best placement of sensors, there are 2n possible configurations. Extensive research has been conducted on the subject of the traffic sensor location problem (TSLP), but limited emphasis has been placed on the associated surveillance issues. The SCLP conducts an investigation to determine the optimal placement and quantity of a specific type of traffic sensor to achieve a pre-established target. The researchers Owais, Mahmoud, and Ahmed I. Shahin conducted a study whereby they devised both exact and heuristic algorithms to address the stochastic capacitated location problem (SCLP) as a subproblem of the transportation and location problem (TSLP). These algorithms were specifically designed to handle large-scale transportation networks [20]. The methodology presented for solving the problem is based on the utilization of random priority selection metaheuristics in conjunction with a column-generating mechanism based on the shortest

path algorithm. The proposed framework is the sole method for implementing the precise method as it effectively avoids the need to exhaustively enumerate all potential paths within a network.

Transportation networks are crucial in facilitating the identification of optimal routes between pairs of origin and destination points. Operators conduct research on this issue in order to optimize network performance, whereas consumers want the created routes to provide the most efficient means of transportation from their starting points to their intended destinations. The issue of variability in journey time is commonly addressed by simplifying the assumption that a probability density function (PDF) exists for each link. In practical scenarios, it is difficult to accurately anticipate the travel time of each link due to variations. As a result, algorithms are necessary to determine the most efficient routing, taking into account real-time traffic conditions on all links. Consequently, the outcome is the creation of a Stochastic Transportation Network (STN) which produces a collection of routes for every pair of nodes (origin/destination). The path sets are designed from the user's perspective, taking into account that this routing does not affect network conditions. A new pre-routing methodology for stochastic transportation networks is introduced in this study. The aim of this approach is to evaluate a predetermined set of pre-generated paths [21] through the simulation of demand fluctuations inside the network during a specified analysis time frame. The methodology used in this study reveals the anticipated path durations and the dependability of each path for the networks under examination. A probability distribution is derived from a dataset consisting of 24,000 generations in order to simulate various network traffic loading scenarios.

### 4.1. Algorithm of the Whole System

The decision issue is how to carry out transport tasks quickly. In addition, the shuttles must avoid each other during the operation process. The feasible routes that the shuttle uses for delivering goods can be considered a closed circle, and the shuttles can only move in two directions. Therefore, optimizing the system in terms of running time and power consumption is a very difficult task. Considering the consumed energy, it is uncertain whether the system can execute a certain task successfully. Moreover, based on the number of stations and the number of shuttles in the whole system, we will require different types of control algorithms to operate all the shuttles.

There are two requirements:

1.  Pre-orders need to be processed as soon as possible to avoid long waiting times for delivery.
2.  The shuttles that need to change batteries should be given priority changing to avoid the situation of running out of batteries, which can become an obstacle on the route. Because of these two conditions, we will establish four rules that match the requirements of the system:

    (a)  Rule 1: If two shuttles meet on the route, the shuttle with the lower priority must avoid the other.

    (b)  Rule 2: Each shuttle can only carry out one task at a time. Shuttles must go through four stages on each mission. The task must go through four stages from the request to the completion of delivery. Shuttles approaching mission completion will be selected for the task.

    (c)  Rule 3: When operating the entire system, it goes through two processes:

      •  Process 1: Change the state of the shuttle and upload completed missions to MySQL;
      •  Process 2: Find new locations.

    (d)  Rule 4: The system's shuttles will be reset before it enters another new process. There are seven statuses for the shuttles, and each has a different priority level. The algorithm will determine the next location for each shuttle based on the status's priority. The priority for each status is as follows.

The initial status of shuttles will define in Table 2. The system will determine the travel routes for the shuttles from the current time until it changes its status. We will show an overview of how the algorithm works in Figure 7 and Table 3. We are still contending that the energy consumed by the shuttle when moving simultaneously is the same.

**Table 2.** Status breakdown table of shuttles.

| Priority | Status of Shuttles | Operation |
|---|---|---|
| 1 | The changing battery shuttle | The shuttle is changing the battery |
| 2 | The stopping shuttle | The shuttle is loading or unloading in the station |
| 3 | The nearing shuttle | The shuttle had just gone to its station but could not get into |
| 4 | The needed-changing battery shuttle | The shuttle has a battery which is lower than the "safe battery" level (the amount of energy the shuttle needs to require a battery change) |
| 6 | The delivery shuttle | The shuttle has goods on it and is delivering the goods |
| 7 | The free shuttle | The shuttle does not have goods on it and is ready for delivery |

As shown in Figures 8–10, there are only two missions to undertake, namely, "Delivery at Station 2" and "Take goods at Station 1". The figures and tables demonstrate how shuttles are chosen to carry out the work. If shuttle B is chosen to perform all missions in 0 s, it would save energy for the entire system. However, the algorithm cannot select it because it simply calculates until the system changes. From 0 to 5 s, the state of the system is the same as the one shown in Table 3; the algorithm will calculate the position of the shuttles during that time, but after 5 s, the algorithm will not be able to calculate because the state of the system has changed into the states shown in Table 4.

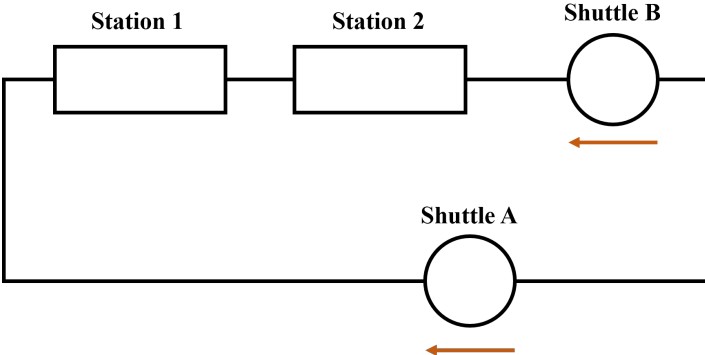

**Figure 8.** The position of the shuttle at the time 0 s.

**Table 3.** Data at 0 s.

| Priority | Shuttle | Time to End Mission | State of Shuttles | Mission |
|---|---|---|---|---|
| 1 | B | 5 s | Delivery shuttle | Delivery in Station 2 |
| 2 | A | 80 s | Free shuttle | Take goods at Station 1 |

**Table 4.** Data at 5 s.

| Priority | Shuttle | Time to End Mission | State of Shuttles | Mission |
|---|---|---|---|---|
| 1 | B | 40 s | Stopping shuttle | Unload in Station 2 and return |
| 2 | A | 75 s | Free shuttle | Take goods at Station 1 |

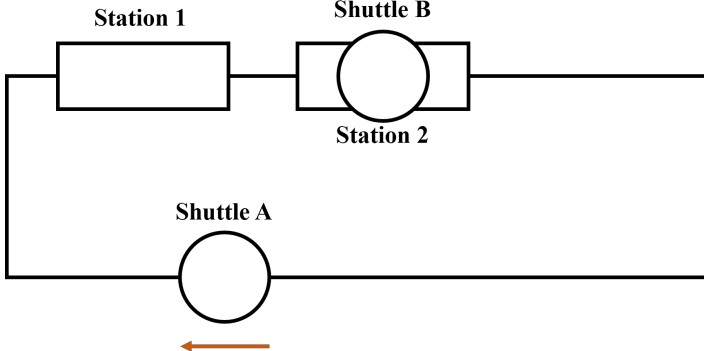

**Figure 9.** The position of the shuttle at the time 5 s.

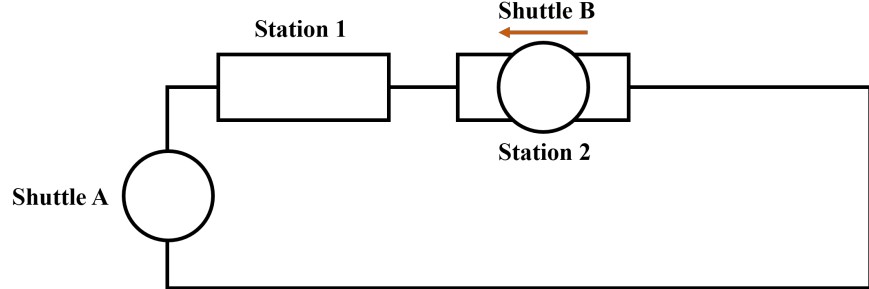

**Figure 10.** The position of the shuttle at the time 45 s.

In any state of the system, the algorithm always tries to choose the better energy-saving solution. As shown in Figure 10 and Table 5, at the time 45 s, the system's state changes; Shuttle B is free, and the distance from B to station 1 is closer than that of shuttle A. As a result, Shuttle B is selected to get the goods, and Shuttle A is stopped. The control algorithm of the system is presented in the form of pseudocode Algorithm 1 and the flow chart in Figure 11.

**Table 5.** Data at 45 s.

| Priority | Shuttle | Time to End Mission | State of Shuttles | Mission |
|---|---|---|---|---|
| 1 | B | 20 s | Free shuttle | Take the goods at Station 1 |
| 2 | A | 0 s | Free shuttle | Does not have mission |

---

**Algorithm 1** Control Algorithm

1: Load the location, state of the shuttles and the stations, and the mission from MySQL;
2: Change the state of the shuttles, and the stations and update the required mission to MySQL;
3: Find a new location for the shuttles;
4: Upload the control signal to MySQL and to the shuttles, the stations execute and those will again reply with the new position and new status to MySQL;
5: **if** Still have a mission to do on MySQL **then**
6:     return to step 1;
7: **else**
8: **end if**

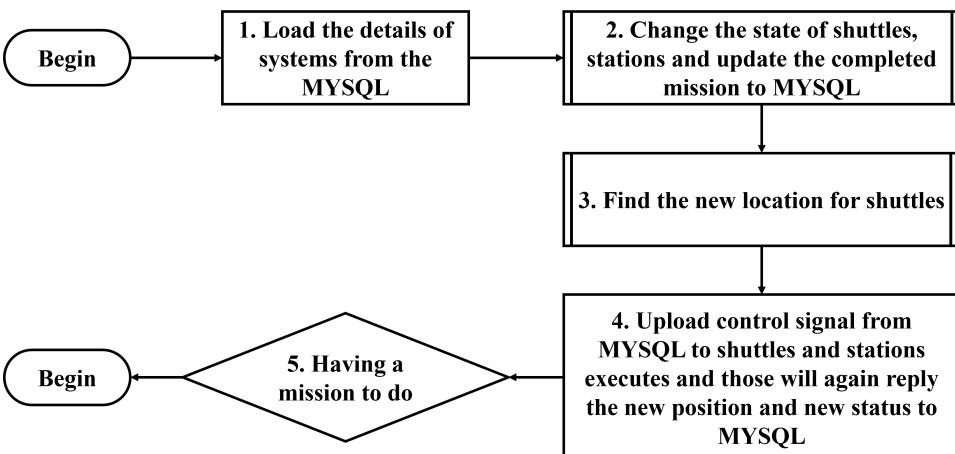

**Figure 11.** Algorithm flowchart of the whole system.

*4.2. Load Data of the System from Mysql*

Before starting the whole process, the PC will connect directly to the data in MYSQL and get the information about the shuttles and the stations.

*4.3. Change the States of the Shuttles*

Based on the tasks of the shuttles, this process will determine which shuttle needs to be prioritized. For example, if one shuttle has just arrived at its station, it will unload the goods first, changing from the group/state "the delivery shuttle" to "the stopping shuttle".

The state of the system depends on the shuttles' tasks and the operation process. The operation process of the whole system is as follows in Algorithm 2.

---

**Algorithm 2** Change the states of the shuttles

---

**Require:** Check the battery level and change the state of the shuttles still on the route with the battery is lower the safe battery
  1: Check the battery level and change the state of the shuttles still on the route
  2: Find the closest Station to change the battery
  3: Get the shuttle need to charge into its changing station
  4: Do the unloading
  5: Find the right shuttle to get the goods
  6: Do the loading
  7: Get the shuttles to Stations to avoid the higher priority shuttles
  8: Get the shuttles out from Station
**Ensure:** Change the state of the Shuttles and Stations and update the completed missions to MySQL

---

*4.4. Find a New Location for the Shuttles*

4.4.1. The Object Function of the System

As the objective function of the system, we define the following:

$\Delta T_{\text{maximum}}$ is the maximum time that the system can calculate. $\Delta T_{\text{maximum}}$ depends on the complexity of algorithms, the computer capacity, and the skill of coders.

$E_n^0(t_0)$ is the lowest energy cost at time $t_0$ for Shuttle $n$th to complete its mission.

By convention, energy is expended only when the Shuttle is moving and is equal in all states (ignore the starting energy of the motor and consider that the energy consumed by the shuttle when there is no cargo is equal to the energy the vehicle expends when it is loaded). As a result, we have the following:

$$E_n^0(t_0) = C \cdot \frac{\text{Shortest distance from Shuttle } n\text{th to complete its task at time } t_0}{v_i,}$$

where $C$ is a constance and $v_i$ is the speed of that shuttle.

$E_n^0(t_0)$ is the energy which the $n$th Shuttle will consume to complete its task from the time $t_0$.

$E(t_0)$ is the total energy of the shuttle system needed to complete its task from time $t_0$, so $E(t_0) = \sum E_n(t_0)$, where $n$ is any shuttle in the system.

Based on the definition of $E_n(t_0)$, the $i$th shuttle has the status "the changing battery shuttle" or "the stopping shuttle", and the $E_i(t_0)$ is always 0 because that shuttle just needs to stand still in order to complete its mission at the station. Another case is "the free shuttle": the shuttle in this group that is not chosen to acquire goods will not have a task to complete so the energy using for completing its task is 0. Therefore, we have another definition of $E(t_0)$:

$E(t_0) = \sum E_m(t_0)$ is $m$, which is the shuttle with the task needing to be run.

### 4.4.2. A Few Notes to Take before Implementing

As previously mentioned in the algorithm section, there are two factors that can be optimized: time and energy. Therefore, the algorithm will prioritize energy optimization during that time. In order to optimize energy, the total travel distances of the shuttle system must be the shortest route for each one. Regarding these, there are many algorithms that we can apply, such as Dijkstra, Bellman-Ford, and A* algorithms, etc. However, the algorithms, as mentioned above, can only solve the problem of finding the shortest path for each shuttle, not for the entire system. In our case, the operation of all the shuttles of the system must be considered during the operation time of the whole system. Two shuttles can be at the same station at the same time in each algorithm mentioned earlier. In our situation, it is necessary to have either a shuttle or no shuttle at a station. So, the idea of the algorithm, which is based on the position of the shuttles that will be proposed in the future, is to decide whether the shuttles will be stuck or not.

In Figure 12 and the detail of explanation in Table 6, Shuttle B is moving in the shortest direction to deliver goods at Station 1 (the most energy-saving route for Shuttle B), and Shuttle A is also moving in the shortest direction to pick up goods at Station 2 (the most energy-saving route of shuttle A). The shortest route is through these operations.

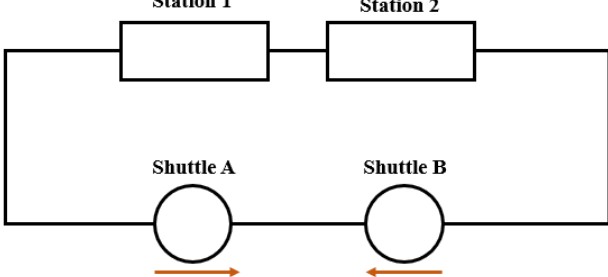

**Figure 12.** Two shuttles will cause jam situation when traveling as marked direction.

If two shuttles continue to move forward, there will be an accident. The solution to this problem is for two shuttles to recognize that they will be stuck if they continue to proceed in this manner. The situation will result in one shuttle giving way to the other. However, if "Station 1" is in the middle of the two shuttles and Shuttle B can come to "Station 1" before colliding with Shuttle A, this situation will be named "no stuck".

**Table 6.** The details of two shuttles.

| Priority | Shuttle | State of Shuttles | Mission |
|---|---|---|---|
| 1 | 1 | Delivery shuttle | Delivery at Station 1 |
| 2 | 2 | Free shuttle | Get good at Station 2 |

### 4.4.3. A Method for Finding a Solution

As for the logical mathematics, the system is located in hospitals to deliver drugs or items from different warehouses to customers. The processing of orders should be prioritized according to first-in-first-out approach. The shuttles with higher priority will be given priority to choose the travel route first because they are fulfilling the required orders first. The algorithm will calculate the $t_0$:

Step 1: Find the shortest ways for the shuttles to complete their tasks. The rail system can be considered a closed circle. The shortest route from A to B can be found by simply examining which AB arc is shorter.

Step 2: From the highest priority to the lowest priority, the shuttle should proceed to choose its route according to the rule that the lower priority shuttle must check the route of the higer priority chuttles and choose its route so as to make the $E_i(t_0)$ as minimal as possible or, more precisely, to save most the battery. To reduce the complexity of the algorithm, we decided that the shuttle with the highest priority is always guaranteed to take the shortest distance at all times.

Figure 13 and Table 7 show the directions taken by shuttles I, II and III to complete their tasks.

**Table 7.** The mission and priority of shuttles I, II, and III.

| Priority | Status of Shuttles | Operation |
|---|---|---|
| 1 | I | To deliver at Staion 1 |
| 2 | II | To deliver at Station 2 |
| 3 | III | To acquire goods at Station 3 |

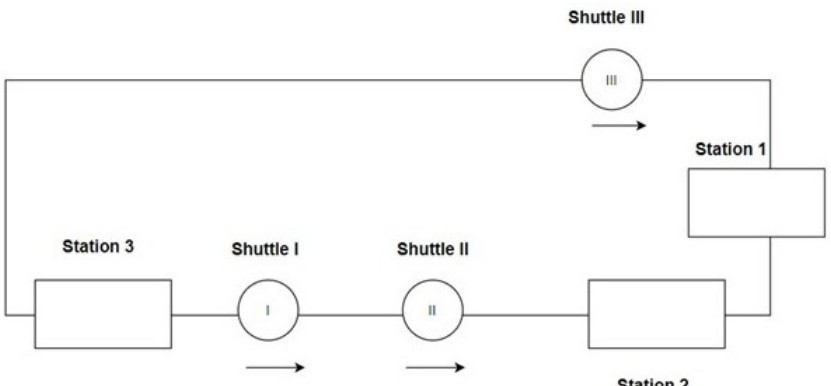

**Figure 13.** Illustration of the position of three shuttles.

At that moment, the energy which will be consumed to complete three tasks is $E$. The system in Figure 12 includes just three shuttles, so $E(t_0) = E_I(t_0) + E_{II}(t_0) + E_{III}(t_0)$.

$E_I^0$, $E_{II}^0$, and $E_{III}^0$ denote the least amount of energy consumed for shuttles I, II, III to complete their tasks.

$$\begin{cases} E_I^0(t_0) = C.\frac{\text{Shortest distance from Shuttle I to Station 1}}{v_I} \\ E_{II}^0(t_0) = C.\frac{\text{Shortest distance from Shuttle II to Station 2}}{v_{II}} \\ E_{III}^0(t_0) = C.\frac{\text{Shortest distance from Shuttle III to Station 3}}{v_{III}} \end{cases}$$

The system is currently set up such that the cars always adjust at the same speed, so $v_I = v_{II} = v_{III} = v_0$.

If Shuttle A or B decides to run in a direction other than the shortest, the $E_I(t_0) \geq E_I^0(t_0)$, $E_{II}(t_0) \geq E_{II}^0(t_0)$, $E_{III}(t_0) \geq E_{III}^0(t_0)$. So $E(t_0) \geq E_I^0(t_0) + E_{II}^0(t_0) + E_{III}^0(t_0)$.

The "=" appears when $E_I(t_0) = E_I^0(t_0)$, $E_{II}(t_0) = E_{II}^0(t_0)$, $E_{III}(t_0) = E_{III}^0(t_0)$. This means the shuttles I, II and III have just two options:

1. Keep running and allow for the shortest direction;
2. Remain standing.

For here, from the highest priority to the lowest priority, the Shuttle I chooses the direction first. Next to it is the Shuttle II, and the last is the Shuttle III. Moreover, Shuttle I is the highest priority shuttle, so its route is the shortest route to complete its task.

In Figure 14, the $\Delta T_{\mathrm{maximum}}$ is 7, and the shortest route to end the task of Shuttle I comes from A- > B- > C- > - > H, which is the chosen route of the Shuttle, so $E_I(t_0) = E_I^0(t_0)$. Next is Shuttle II. It must compare its shortest route with the route of Shuttle I (the shuttle with the higher priority). From time 0 to time 7, there are instances of "no stuck" occuring between Shuttle II and I, so the decision of Shuttle II is to run and the route of Shuttle II is setup: $E_{II}(t_0) = E_{II}^0(t_0)$. The last is Shuttle III. It must check the route of Shuttle I and II. However, between time 5 and time 6, the shortest route of Shuttle III coincides with the route of Shuttle 1. So, to make $E_{III}(t_0) = E_{III}^0(t_0)$, Shuttle III's only option is to remain standing.

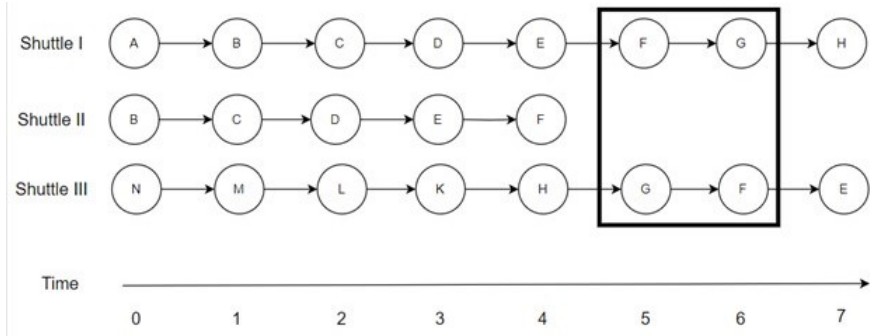

**Figure 14.** The shortest route to complete the tasks of each Shuttle (I, II, and III).

Finding the station to avoid: The majority of the freight process occurs in situations where the shuttles meet without any other stations to avoid. The reason for this is that the number of main stations and substations required to avoid each other is very small and the actual travel distance is very large (maybe 5–10 km). The size of the stations is about 0.5 m, but the number of stations can be less than 100. The distance traveled is less than 1% of the total size of the stations.

But the fact that there are bypass stations in the middle of the road is still at issue. The above example is carried forward with the additional assumption that Station 4 is located in the middle of the Shuttle III route, as shown in Figure 15. The information concerning shuttles I, II, and III is still same as the Table 7.

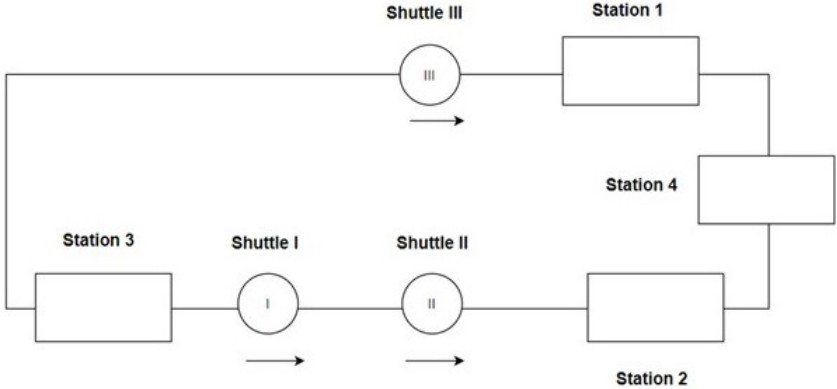

**Figure 15.** Station 4 in the route of Shuttle III.

According to Figures 14 and 15, Station 4 is on the shortest route of Shuttle III. There are two cases that could unfold from here.

Case 1: Station 4 is on the shortest route of Shuttle III after the time 5. Choosing a station that is not selected near the shuttle has a higher priority in general. If this is the case, it can be divided into three options.

+Option 1: Shuttle I will run to avoid in Station 4. Generally, the higher priority shuttle will enter the station for avoidance. This option can make $E(t_0) = E^0(t_0)$. Because this option does not change the route of the shuttle with a higher priority, it only prolongs the quest completion time of that shuttle.

+Option 2: Shuttle III will run to avoid in Station 4. In general, shuttles with lower priority will enter the station to avoid Station 4. This option makes $E(t_0) = E^0(t_0)$. However, this option causes the higher priority shuttle to wait until the lower priority shuttle enters the station.

+Option 3: The shuttle with lower priority will wait while the higher priority shuttle runs. This option can make $E(t_0) = E^0(t_0)$.

All three options do not reduce the value of the function E at the time of consideration. However, options 1 and 2 violate the principle that orders are requested first and processed later, causing some orders to be backlogged. Hence, we chose the option 3 in this case.

Case 2: Station 4 is on the shortest route of Shuttle III before the time 5. In general, choosing the station that has not been selected near the shuttle has lower priority. In this case, it is obvious to choose that the shuttle with the lower priority to come into Station 4 to avoid other shuttles. In general, the shuttle with lower priority goes into that station to avoid other shuttles. This option does not change any of the shortest routes of any of the shuttles. It just takes a longer time to complete the tasks of shuttles with lower priority. However, it can be accepted because it does not violate anything.

Finding the $\Delta T_{\text{maximum}}$: In another example, we have the same case but $\Delta T_{\text{maximum}}$ is 5, which means the system can check from time 0 to time 5. The "stuck" parameter cannot be discerned because the maximum calculation is time 5, so Shuttle 3 is set up to run. Therefore, the $\Delta T_{\text{maximum}}$ is very important. However, in this paper, we have not found a specific way to calculate $\Delta T_{\text{maximum}}$. We only select $\Delta T_{\text{maximum}}$ according to the amount of time taken for the highest priority shuttle to complete its task. Because it only takes one shuttle to complete its task, the entire state of the shuttle system will change, so the system only needs to calculate until one shuttle completes its task; on the other hand, when no shuttle has completed its task, there will be a few other random factors that affect the state of the system, such as a new order being added by the user, or the shuttle being in motion, or the battery indicator sudden weakening. Calculating the time required to complete the task of a single shuttle is a challenging task. The time required to complete the task of the highest-priority shuttle is always calculated by convention.

Definitions and Assumptions

- "Had-run" is the list of shuttles that were set direction and priorities;
- "Distance safe" is the minimum distance between two shuttles when traveling on the route;
- "The wait shuttle" is the shuttle that is about to go into its station to complete the stage of its mission but cannot because there is a shuttle that is unloading, loading, or changing the battery in the station.
  To allow the system to understand two meanings, "stuck" and "no stuck", the algorithm is designed as follows.
  Based on the above principle, the Algorithm 3 "Find a new location" performs the following pseudocode.

**Algorithm 3** Find the new location

1: **Begin**
2: Create the "had-run" list
3: Let the shuttles in front of the Station having shuttles want to return the route go away
   ▷ Step 1
4: Find the direction for the shuttles                                    ▷ Step 2
5: Check the distance between two shuttles                                ▷ Step 3
6: **End**

1. Let the shuttles in front of the station with the shuttles wanting to return to the route go away with the Algorithm 4.

   The Table 8 is the task of all shuttles When Shuttle A is returning to the old route, shuttles B and C should give way. Shuttle C must turn left on the left side of the station, and shuttle B must turn right on the right side of the station. However, when Shuttle C turns left, Shuttle C can collide with Shuttle D as the distance between the two shuttles is too short. However, shuttles B and E cannot run into each other because the distance is far. In Figure 16, when shuttle C turns left, if the distance between C and D is shorter than the "safe distance", Shuttle D must follow Shuttle C. After that, we check the next shuttle on the left side of Shuttle D in order that there will exist two groups of shuttles to the left and right of the station. When the first shuttle of the group starts to run, the other shuttle must follow. All shuttles are set as the highest priority.

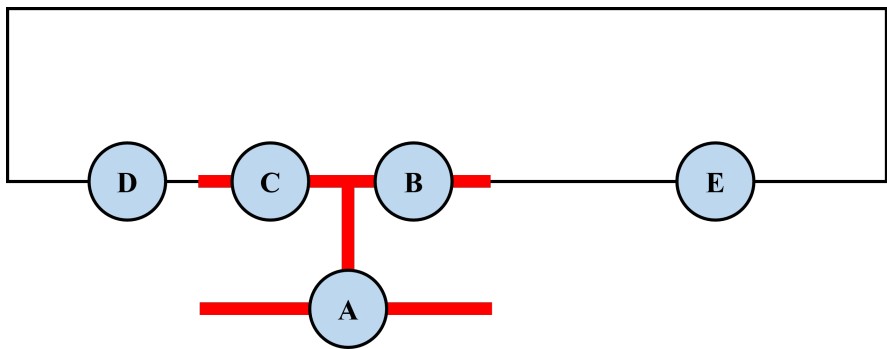

**Figure 16.** Shuttle A wants to return to the old route but shuttles B and C are standing in front of the station.

**Table 8.** The details of all shuttles after "Make the shuttle stand in front of the Station".

| Shuttles | Priority | Direction |
|---|---|---|
| B | 1 | Counter-clockwise |
| C | 1 | Clockwise |
| D | 1 | Clockwise |

**Algorithm 4** Make the shuttle standing in front of the Station

1: **Begin**
2: Find the closest shuttle on each side of the Station                   ▷ Step 1
3: Find the group on each side                                            ▷ Step 2
4: Check each group need to run?                                          ▷ Step 3
5: **if** that still have a shuttle of the group standing in front of Station **then**
6:     All shuttles in group run are set the highest priority;
7: **else**
8:     Do not do anything
9: **end if**
10: **End**

2. Find the moving direction of the shuttles is shown in Algorithm 5: After the process and the priorities of the shuttles have been completed, all that remains is to find a way for the shuttles to carry out their missions, assuming the speed and the energy used by the shuttles are always the same. The control algorithm for this problem is introduced as Figure 17 and the details of the system before changing status to "Find the direction of the system" at the time $t_0$ is shown in Table 9.

---

**Algorithm 5** Find the direction for each shuttles

---

1: **Begin** Find the direction for each shuttle:
2: **for** (CHECK-shuttle) = Highest priority shuttle: Lowest priority shuttle **do**
3:     **if** CHECK-shuttle had been set the direction **then**
4:         Change to the next priority shuttle;
5:     **else**
6:         Step 1: Change to the next priority shuttle;
7:         Step 2: Create a group of shuttles to move in the nearest direction found in Step 1;
8:         *Rules for creating groups:*
9:         Step 1:
10:         Set the (last-shuttle) = CHECK-shuttle
11:         Add the CHECK-shuttle into "group"
12:         Step 2:
13:         **if** distance between last-shuttle and the shuttle next to it $\leq$ "safe distance" **then**
14:             Add the shuttle which is next to the last shuttle into "group";
15:             Set (last-shuttle) = the shuttle has just been added
16:         **else**
17:             End find the "group"
18:         **end if**
19:         Step 3: Check the "stuck" with the higher priority shuttles if the group goes in the intended direction
20:         **if** "no stuck" **then**
21:             Set the direction of all shuttles in the group in the same as the direction found in Step 1;
22:             Set the priority of all shuttles in the group as the same as the priority of the CHECK-shuttle;
23:             ADD all shuttles in group into "had-run" list;
24:             Change to check the next priority shuttle;
25:         **else**
26:             **if** (Have a station which was not be selected before coming to the jam position) **then**
27:                 Set that Station will be used for CHECK-shuttle avoiding;
28:                 Set the direction of all shuttles in the group in the same as the direction found in Step 1;
29:                 Set the priority of all shuttles in the group as the same as the priority of the CHECK-shuttle;
30:                 Change to check the next priority shuttle;
31:             **else**
32:                 Set direction of CHECK-shuttle is "non-run";
33:                 ADD all CHECK-shuttle in the group into "had-run" list;
34:                 Change to check the next priority shuttle;
35:             **end if**
36:         **end if**
37:     **end if**
38: **end for**
39: **End**

---

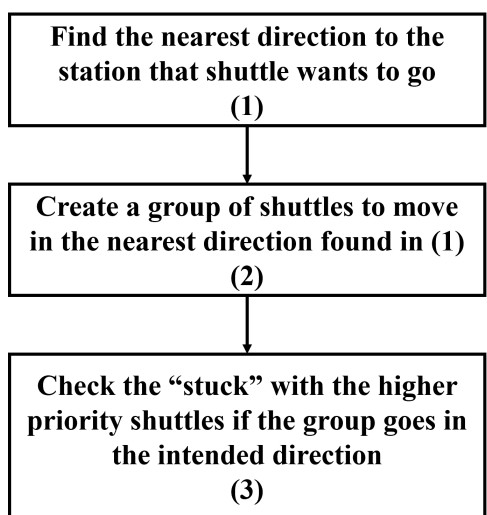

**Figure 17.** The flow chart for "Find the direction for each shuttles".

**Table 9.** Details of the system before changing status to "Find the direction of the system" at the time $t_0$.

| Priority | Shuttle | State of Shuttle | Mission | Direction |
|---|---|---|---|---|
| 1 | 1 | Need charge shuttle | Go charge at Station A | Not set |
| 2 | 2 | Delivery shuttle | Delivery at Station D | Not set |
| 3 | 3 | Delivery shuttle | Delivery at Station F | Not set |
| 4 | 4 | Free shuttle | Get good at Station E | Not set |
| 5 | 5 | Free shuttle | Get good at Station B | Not set |
| 6 | 6 | Free shuttle | Don't have mission | Not set |
| 7 | 7 | Free shuttle | Don't have mission | Not set |

Figure 18 illustrates all the possible situations that the delivery system can encounter. Table 10 is the entire information of the shuttles. Here is how the algorithm solves the problem: we apply the above algorithm as introduced in Figure 18 with the operation condition as introduced in Table 10. With every cycle, the system will search for the travel direction for the shuttle that has not yet set the highest priority.

First turn: Shuttle 1 with the highest priority that has not been set in the travel direction.

Step 1: The nearest direction to Station A is clockwise;

Step 2: Follow the clockwise direction, the next shuttle is shuttle 6, but;

Step 3: If the "group of Shuttle 1" moves, the shuttles will not collide with each other because there are no shuttles with higher priority than Shuttle 1. So, Shuttle 1's direction is set to clockwise.

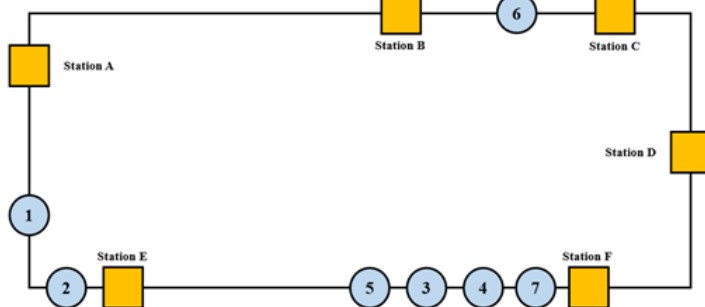

**Figure 18.** The sytems at the time $t_0$.

**Table 10.** Details of all the shuttles after the first turn.

| Priority | Shuttle | State of Shuttle | Mission | Direction |
|---|---|---|---|---|
| 1 | 1 | Need charge shuttle | Go charge at Station A | Clockwise |
| 2 | 2 | Delivery shuttle | Delivery at Station D | Not set |
| 3 | 3 | Delivery shuttle | Delivery at Station F | Not set |
| 4 | 4 | Free shuttle | Get good at Station E | Not set |
| 5 | 5 | Free shuttle | Get good at Station B | Not set |
| 6 | 6 | Free shuttle | Don't have mission | Not set |
| 7 | 7 | Free shuttle | Don't have mission | Not set |

Second turn: Shuttle 2 has not been set the travel direction and it has the highest priority among the remaining shuttles and the details of all the shuttles after the second turn are shown in the Table 11

Step 1: The nearest direction to D is counter-clockwise;

Step 2: Following the counter-clockwise direction, the next shuttle is Shuttle 3 but the distance between Shuttle 3 and Shuttle 2 is bigger than "safe distance", so the "group of Shuttle 2" has Shuttle 2 only;

Step 3: If "group of Shuttle 2" moves, the shuttles will not collide with each other. Shuttle 1 is of the higher priority and it can collide with Shuttle 2 if it still moves in clockwise-direction. However, it will finish its work before meeting Shuttle 1. So, Shuttle 2's direction is set to counter-clockwise.

**Table 11.** Details of all the shuttles after the second turn.

| Priority | Shuttle | State of Shuttle | Mission | Direction |
|---|---|---|---|---|
| 1 | 1 | Need charge shuttle | Go charge at Station A | Clockwise |
| 2 | 2 | Delivery shuttle | Delivery at Station D | Counter-clockwise |
| 3 | 3 | Delivery shuttle | Delivery at Station F | Not set |
| 4 | 4 | Free shuttle | Get good at Station E | Not set |
| 5 | 5 | Free shuttle | Get good at Station B | Not set |
| 6 | 6 | Free shuttle | Don't have mission | Not set |
| 7 | 7 | Free shuttle | Don't have mission | Not set |

Third turn: Shuttle 3 has not been set the travel direction and it has the highest priority among the remaining shuttles and the details of all the shuttles after the third turn are shown in the Table 12

Step 1: The nearest direction to F is counter-clockwise.

Step 2: Following the counter-clockwise direction, the next shuttle is Shuttle 4 and the distance between Shuttle 3 and Shuttle 4 is smaller than the "safe distance", so the "group of Shuttle 3" has Shuttle 4. Next to Shuttle 4 is Shuttle 7, and the distance between them is smaller than the "safe distance"; therefore, Shuttle 7 is added into "group of Shuttle 3". The shuttle next to Shuttle 7 is Shuttle 6 but the distance between them is bigger than the "safe distance" so it is not added in the "group of shuttle 3". "Group of Shuttle 3" include shuttles 3, 4 and 7.

Step 3: If "group of Shuttle 3" moves, the shuttles will not collide with each other. Shuttle 2 has the same direction with "group of Shuttle 3" so it will not "stuck" with "group of Shuttle 3". Shuttle 1 has the opposite direction, but it will finish before running into "group of Shuttle 3". So all the shuttles in "group of Shuttle 3" are set to clockwise, and the priorities are the same as that of Shuttle 3.

**Table 12.** Details of all the shuttles after the third turn.

| Priority | Shuttle | State of Shuttle | Mission | Direction |
|---|---|---|---|---|
| 1 | 1 | Need charge shuttle | Go charge at Station A | Clockwise |
| 2 | 2 | Delivery shuttle | Delivery at Station D | Counter-clockwise |
| 3 | 3 | Delivery shuttle | Delivery at Station F | Counter-clockwise |
| 3 | 4 | Free shuttle | Get good at Station E | Counter-clockwise |
| 3 | 7 | Free shuttle | Don't have mission | Counter-clockwise |
| 5 | 5 | Free shuttle | Get goods at Station B | Not set |
| 6 | 6 | Free shuttle | Don't have mission | Not set |

Fourth turn: Shuttle 5 has not been set the travel direction and it has the highest priority among the remaining shuttles and the details of all the shuttles after the fourth turn are shown in the Table 13

Step 1: The nearest direction to B is clockwise.

Step 2: Following the clockwise direction, the next shuttle is Shuttle 6, but the distance between Shuttle 5 and Shuttle 6 is bigger than the "safe distance". So, "Group of shuttle 5" has Shuttle 5 only.

Step 3: If "group of shuttle 5" moves, the shuttle will collide with those of "group of Shuttle 2". So the algorithm calculates the locations which cause the "stuck" between the two groups is the "O", as shown in the Figure 19.

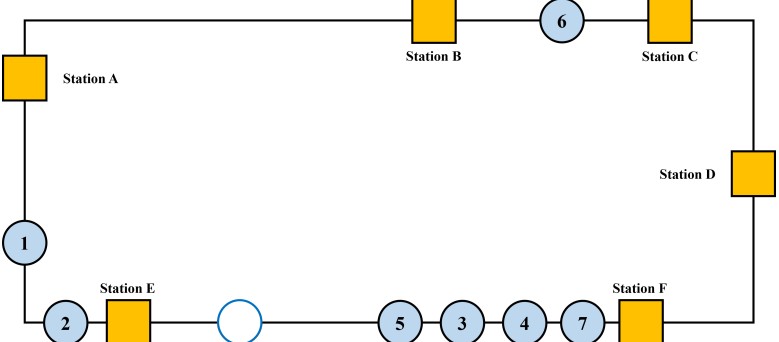

**Figure 19.** The "stuck" happens at "O" position if Shuttle 5 is directed to move in a counter-clockwise direction.

From the location of Shuttle 5 to the "O" position, there is no free station, so the "group of shuttle 5" is set the "non-run" direction.

**Table 13.** Details of all the shuttles after the fourth turn.

| Priority | Shuttle | State of Shuttle | Mission | Direction |
|---|---|---|---|---|
| 1 | 1 | Need charge shuttle | Go charge at Station A | Clockwise |
| 2 | 2 | Delivery shuttle | Delivery at Station D | Counter-clockwise |
| 3 | 3 | Delivery shuttle | Delivery at Station F | Counter-clockwise |
| 3 | 4 | Free shuttle | Get good at Station E | Counter-clockwise |
| 3 | 7 | Free shuttle | Don't have mission | Counter-clockwise |
| 5 | 5 | Free shuttle | Get goods at Station B | Non-run |
| 6 | 6 | Free shuttle | Don't have mission | Not set |

Fifth turn: Shuttle 6, its direction is "non-run" and the details of all the shuttles after the fifth turn are shown in the Table 14.

**Table 14.** Details of all the shuttles after the fifth turn.

| Priority | Shuttle | State of Shuttle | Mission | Direction |
|---|---|---|---|---|
| 1 | 1 | Need charge shuttle | Go charge at Station A | Clockwise |
| 2 | 2 | Delivery shuttle | Delivery at Station D | Counter-clockwise |
| 3 | 3 | Delivery shuttle | Delivery at Station F | Counter-clockwise |
| 3 | 4 | Free shuttle | Get good at Station E | Counter-clockwise |
| 3 | 7 | Free shuttle | Don't have mission | Counter-clockwise |
| 5 | 5 | Free shuttle | Get goods at Station B | Non-run |
| 6 | 6 | Free shuttle | Don't have mission | Non-run |

What will take place if a new mission appears while the system is processing a task?

"When the shuttles are performing a task at any given time, a new task appears as the Figure 20 will interfere to the system's execution".

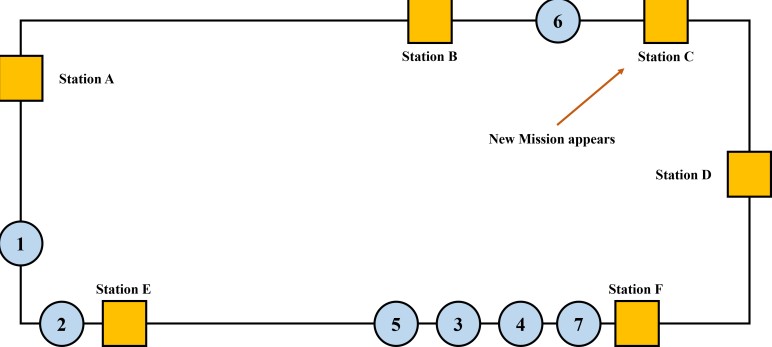

**Figure 20.** A new mission appears while the shuttles are doing their job at the time $t_0$.

The algorithm handles this problem by continuing to compute for the system at the time $t_0$ based on Table 15 and handles a new task in the next computation. The time between two computations is very small, so if there is a loss of energy, it is negligible. In the next calculation, in "Find shuttles to get goods", the system will find the shuttle suitable for this mission if there is "free shuttle".

**Table 15.** Details of the system before having to "find the direction of system" at the time $t_0 + \Delta t$.

| Priority | Shuttle | State of Shuttle | Mission | Direction |
|---|---|---|---|---|
| 1 | 1 | Need charge shuttle | Go charge at Station A | Not set |
| 2 | 2 | Delivery shuttle | Delivery at Station D | Not set |
| 3 | 3 | Delivery shuttle | Delivery at Station F | Not set |
| 4 | 4 | Free shuttle | Get good at Station E | Not set |
| 5 | 5 | Free shuttle | Get goods at Station B | Not set |
| 6 | 6 | Free shuttle | Don't have mission | Not set |
| 7 | 7 | Free shuttle | Don't have mission | Not set |

The new mission will be conducted by Shuttle 6 and the system will find the direction for it at the time $t_0 + \Delta t$.

### 4.4.4. Check the Distances between Two Shuttles and Changing Place

If all the shuttles run according to what the system has set up, an accident can happen. The shuttles are considered as points in the algorithm for ease of calculation. However, they are much larger in size than the point. Therefore, we need to check the distance between two adjacent shuttles and the ones going in the opposite direction to see if there is enough "safe distance moving" or "min-area". If the distance between them is smaller than "min-area", the shuttle with low priority will be labeled as "do not run". However, we need to check if the last shuttle has the same priority in the "had-run" list because it could cause an accident.

*4.5. The Sub-Function Algorithm for Shuttles*

Check the battery level of the shuttles: All of the batteries of the shuttles will be checked at this stage, and if the shuttle has a battery level which is lower than the safety level and the battery that still has enough power for the operation, it will change into the two groups as "the ready charge shuttle" or "the not ready charge shuttle". If that shuttle has the goods, it will be set to "the not ready charge shuttle" and vice versa.

4.5.1. Find the Station for the Shuttles That Need Batteries Changing

This step will set the stations for the shuttles in the groups "battery-changing needed" and "the nearing shuttle" (including shuttles that need to change batteries).

Step 1: Check the shuttles in the group "the nearing shuttle" which have low battery. These shuttles are very near to the station for battery changing, so just set the nearest station for them.

Step 2: Check the shuttles in "need changing the battery shuttle" depending on their priority. Find the nearest stations which are not being used by another shuttle before it.

4.5.2. Let the Shuttles Need to Change Batteries in Their Changing Station

This step will allow shuttles that need to change batteries to stand right outside their changing station. Those shuttles, once they get into the station, will change their status to "The changing battery shuttle". The system will check whether there are other shuttles in the station:

If yes, the shuttle that needs to change battery will be labeled as "The nearing shuttle";

If no, the shuttle will be allowed to go into the station and change status to "The changing battery shuttle".

4.5.3. Unloading Process

In this step, the shuttle standing right outside the unloading station will change status to "The stopping shuttle". The system will check whether there are other shuttles in the station:

If yes, the shuttle will be labeled as "the nearing shuttle";

If no, the shuttle can go into station and change status to "the stopping shuttle".

4.5.4. Find the Shuttles for the Mission

In this step, the system will find suitable shuttles to take the goods for each mission. There are two conditions:

1.　Open the list of missions needing to be completed and sort those depending on the time order;

2.　The loading station of the checking mission that is similar to the loading station of the mission before will not be found and the system will proceed to the next mission

If it is accepted, we begin the steo-by-step process as set out in Algorithm 6.

4.5.5. Loading Process

At this step, the shuttles standing right outside the loading station will be allowed to go into and change status to "The stopping shuttle". The system will check whether there are other shuttles in the station:

If yes, that shuttle will change status to "the nearing shuttle";

If no, that shuttle will go into the station and change its status to "the stopping shuttle".

4.5.6. Get the Shuttles to Stations to Avoid the Higher Priority Shuttles

At this step, the shuttles standing right outside the station will be allowed to take a break in order to give a free route to the higher priority shuttles. The system will check whether there are other shuttles in the station:

If yes, that shuttle will change status to "the nearing shuttle";

If no, that shuttle will be allowed to go into the station.

---

**Algorithm 6** Find the shuttles for the mission

---

1: **Begin** Find the shuttles for missions
2: **for** new-mission = new-mission's first: new-mission's last **do**
3:     **if** Station get goods of new-mission is same as Station get goods of any previous new-missions **then**
4:         Check the next new-mission;
5:     **end if**
6:     **while** Still have free shuttles without work **do**
7:         **if** station get goods have an unloading shuttle or a charging shuttle **then**
8:             Choose that shuttles for that new-mission;
9:         **else**
10:            Find the closest free shuttle which isn't selected and set that shuttles to get the goods of that new-mission;
11:        **end if**
12:    **end while**
13: **end for**
14: **End**

---

### 4.5.7. Get the Shuttles Out from the Station

This step will get the shuttles in the station to return to the line as in Algorithm 7.

---

**Algorithm 7** Get the shuttles out from the station

---

1: **Begin** Get the shuttles out from Station
2: Check shuttles in each station                                          ▷ Step 1
3: **if** that shuttles are "charging shuttle" and the battery which is higher than the battery return **then**
4:     add to "accept get out group";
5: **else if** that shuttles are "stopping shuttle" has completed their loading or unloading job **then**
6:     add to "accept get out group";
7: **else if** that shuttles are in the station because of avoiding the higher priority shuttles pass and it was ordered to get out **then**
8:     add to "accept get out group";
9: **else**
10:    change to next station to continue checking until the last station;
11: **end if**
12: Find the shuttles in front of each station in "the accept get out group"  ▷ Step 2
13: **if** having the shuttles standing in front out the station **then**
14:    set that shuttle wants to get out in that station is "not ready to get out";
15: **else**
16:    set that shuttles want to get out in that station is "ready to get out";
17: **end if**
18: Get the shuttles "ready to get out" out of station                       ▷ Step 3
19: **End**

---

## 5. Simulation Results

The information of the whole system using for simulation as described is presented in Tables 16–18.

**Table 16.** Information on the shuttles.

| Shuttle's Information | Value | Unit |
|---|---|---|
| Quantity of shuttles | 3 | pieces |
| Speed of shuttle | 0.3 | m/s |
| Length of shuttle | 0.3 | m |
| Unloading time | 10 | s |
| Loading time | 10 | s |
| Distance minimum between two shuttles | 0.4 | m |
| Battery consumption when moving continuously | 1.39 | %/s |
| Sample time | 0.1 | s |

**Table 17.** Informations on the stations.

| Station's Information | Value | Unit |
|---|---|---|
| Quantity of stations | 3 | pieces |
| Changing time of Station | 12 | s |
| Length of station | 0.5 | m |
| Distance between 2 rails in station | 0.5 | m |

**Table 18.** Information on the elevators.

| Elevator's Information | Value | Unit |
|---|---|---|
| Quantity of elevator | 2 | pieces |
| Changing time of elevator | 7 | s |
| Length of elevator | 0.5 | m |
| Height between 2 floors | 0.3 | m |

In Figure 21, during one hour, the whole system can successfully handle up to 100 shipments. The relationship between time and the completion of the task is almost a linear function, as the correlation is near 1. In general, the system can be divided into four basic cases:

+ Case 1: When the system has no task, a new task will appear;
+ Case 2: When there is a shuttle processing the mission, there is no order to give way;
+ Case 3: The shuttles of low priority must make way for the ones of high priority;
+ Case 4: The shuttles of low priority must give way to shuttles with higher priority through the use of stations along the way to avoid collision.

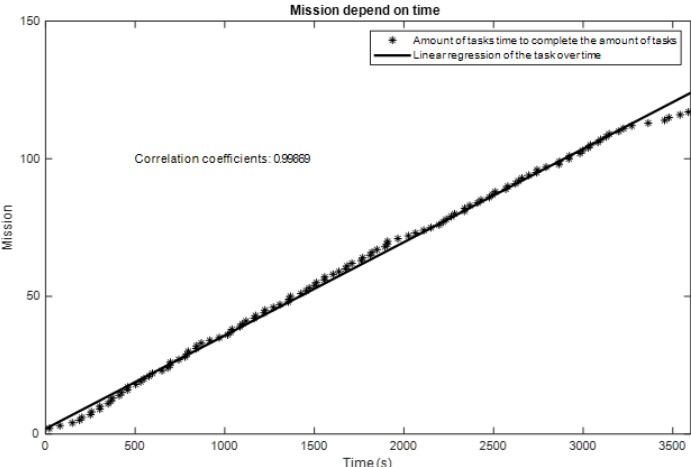

**Figure 21.** The mission was completed on time

In other cases, there will be a combination of the above four basic causes. If all three shuttles have tasks to complete, they will be assigned different priority levels. When any two shuttles can interfere with each other on the road, the system, based on their position and status, will judge if the situation falls within case 2, 3, or 4. To simulate the above four cases and create a control app, we use the C++ programming language and QT programming support tool [22].

### 5.1. Case 1—The System Has Just Begun to Operate

This case appeared right after turning on the transport system. The server system (PC) is being connected by all shuttles, stations, and elevators. In essence, the stations and elevators are the only ones that connect immediately when the system is turned on. Shuttles are currently not activated because they require battery power, and a manager is required to activate them. However, their entire location will be recorded in the database and displayed on the control screen, as shown in Figure 22a. Thus, the administrator can easily trace the activation. Once all devices have been connected successfully. The system is now ready for shipping. The delivery request is entered by the user at this point. The order will be transferred to the database by the system and processed. Figure 22 is an illustration of the case from order receipt to order processing and how the system works. In Figure 22a, the system's state is in a random moment without any mission, i.e., the shuttles can stay anywhere on the route.

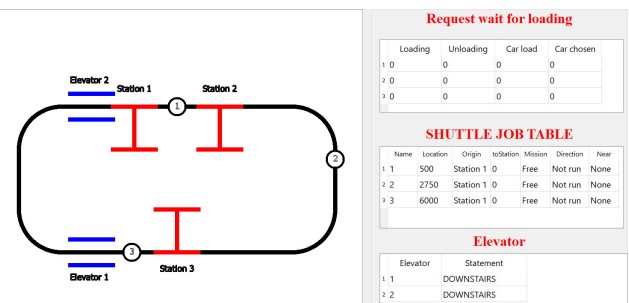

(**a**) The shuttles have no missions.

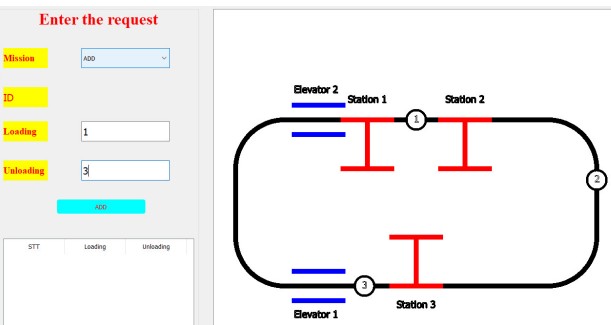

(**b**) The first mission: transporting goods from Station 2 to Station 3,

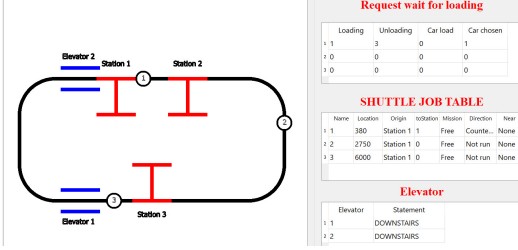

(**c**) The system assigns Shuttle 1 to do the job,

**Figure 22.** *Cont.*

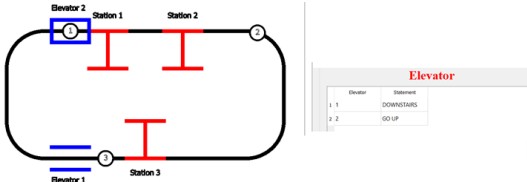

(**d**) Shuttle 1 uses Elevator 2 to go up.

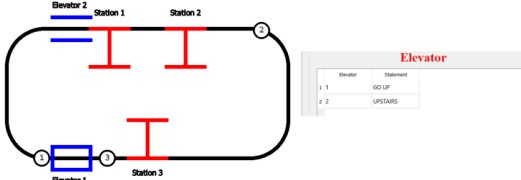

(**e**) Shuttle 1 calls Elevator 1 to go up to pick up the goods.

**Figure 22.** Case 1—The system has just started to operate.

In Figure 22b, a new mission is added: transport goods from Station 1 to Station 3.

In Figure 22c, at the time a mission is added: transport goods from Station 2 to Station 3; the system calculates that Shuttle 1 is closest to Station 2 so it is assigned for that mission. We can see it in the "Request wait for loading" table.

In Figure 22e, Shuttle 1 uses elevator 1 to go up; however, when Shuttle 1 wants to use elevator 1 to go down, the floor of Elevator 1 is different from the floor of Shuttle 1, so Shuttle 1 must call Elevator 1 to go up to acquire its good.

Cases 5.2 to 5.4 show the situations that occur in the transportation system and show how to deal with them.

### 5.2. Case 2—The Shuttle Doing Its Job Must Make Another Shuttle Run Away

This is the case that occurs most frequently in shipping. Currently, shuttles in the transportation process will need to give way to each other so that shuttles of higher priority can complete their missions. Since there were no stations between the shuttles, they were forced to give way. We can imagine that in a situation where two cars are facing each other in a narrow alley, one car has to back up so that the two cars can continue to move. In addition, this situation is shown via Figure 23. In Figure 23a, a new mission appears: Transporting goods from Station 2 to Station 3, and Shuttle 1 is called to load the goods at Station 2.

According to the map in Figure 23b, the shortest distance from Station 2 to Station 3 is clockwise. In this direction, Shuttle 2 is standing in the middle of the road, so Shuttle 1 must catch the group with Shuttle 2 and make the two cars move clockwise.

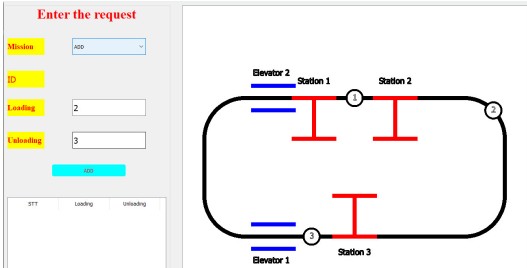

(**a**) A new mission appears: Transporting goods from Station 2 to Station 3.

**Figure 23.** *Cont*.

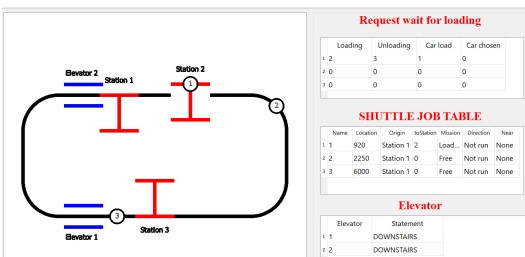

(**b**) Shuttle 1 is chosen to obtain the goods.

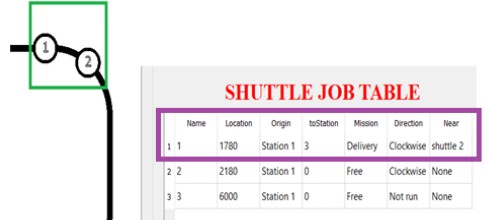

(**c**) Shuttle 1 makes the group, with Shuttle 2 to go.

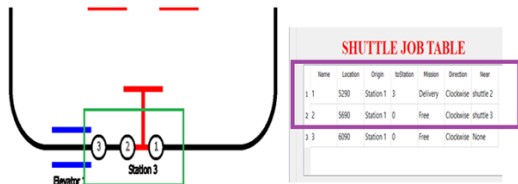

(**d**) The group of Shuttle 1 makes Shuttle 3 go.

**Figure 23.** Case 2—The shuttle doing its job must make another shuttle run away.

### *5.3. Case 3—There Is More than One Shuttle Operating to Carry Out the Mission*

This is exactly the case when two shuttles were predicted by the system to be going against each other long ago. However, between the two shuttles, there is no station to avoid. So, according to the algorithm, the low-priority shuttle decides to stay still and creates a group with the high-priority shuttle so that it can change its priority. This process is shown in Figure 24.

In Figure 24a,b, the mission "transporting goods from Station 2 to Station 3" appears, and Shuttle 1 is chosen to load the goods.

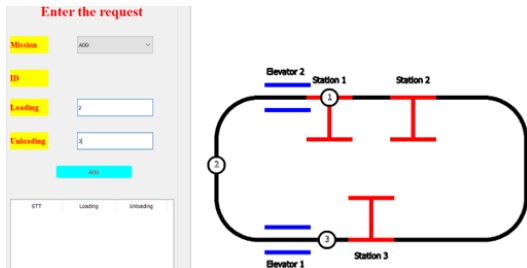

(**a**) A new mission is added: delivering goods from Station 2 to Station 3.

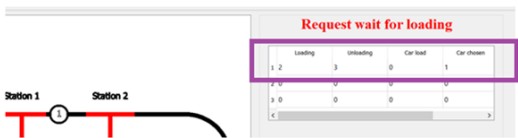

(**b**) Shuttle 1 is selected to load the goods in Station 2.

**Figure 24.** *Cont*.

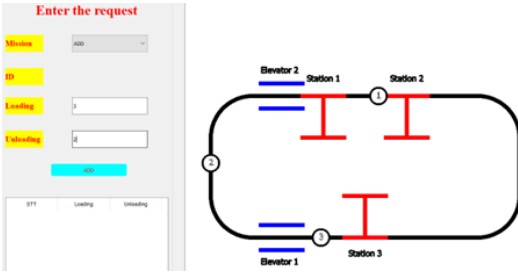

(**c**) A new mission appears.

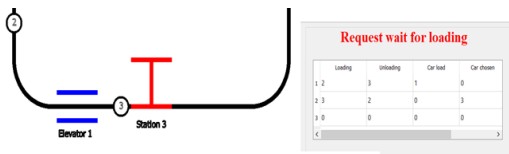

(**d**) Shuttle 3 is chosen for the new mission.

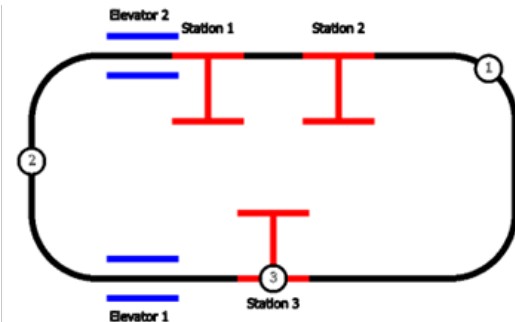

(**e**) Shuttle 3 has just returned to the route.

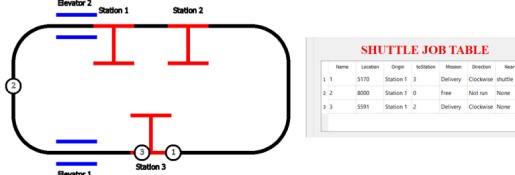

(**f**) Shuttle 3 makes way for Shuttle 1 to finish its mission first.

**Figure 24.** Case 3—There is more than one shuttle operating to carry out the mission.

However, while Shuttle 1 is performing the task, a new mission appears—loading goods from Station 3 to Station 3 (Figure 24c)—and Shuttle 3 is the closest to the station so it is chosen to do the job (Figure 24d).

In Figure 24e, Shuttle 3 returns to the route more slowly than Shuttle 1, so Shuttle 3 has lower priority. Shuttle 3 needs to deliver the goods at Station 2 in a counter-clockwise direction.

However, this option is not possible as Shuttle 1 is moving clockwise towards Station 3. Moreover, Shuttle 3 cannot use any stations between Station 2 and Station 3 to avoid colliding with Shuttle 1. Therefore, Shuttle 3 must stay at Station 3 and wait until Shuttle 1 has completed its mission (as demonstrated in Figure 24f).

*5.4. Case 4—The Shuttle of Low Priority Must Find a Station to Avoid the One of High Priority*

This case is a case where two Shuttles are expected to confront each other after a while. However, between them are stations that can be used to avoid one another. Therefore, they will have the most appropriate calculation. This case has been analyzed in detail in the Mathematics Logics section.

In Figure 25a, the missions of Shuttle 1 and Shuttle 2 are to deliver the goods at Station 1 and Station 3, respectively. Shuttle 1 is given higher priority than Shuttle 2. An accident

can occur if both shuttles deliver the goods simultaneously. In another scenario, Shuttle 2 must make space for Shuttle 1 (as demonstrated in the third case). However, between these two shuttles is Station 2. Shuttle 2, which is near Station 2, can travel there and make way for Shuttle 1.

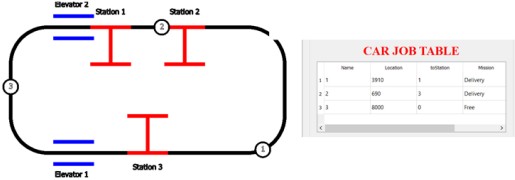

(**a**) The mission and the location of Shuttle 1 and Shuttle 2.

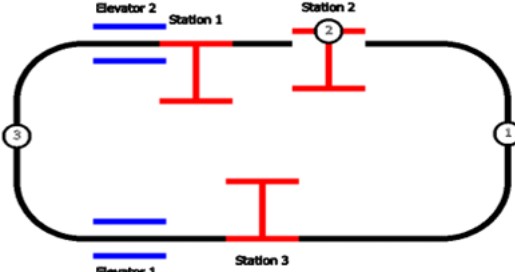

(**b**) Shuttle 2 uses Station 2 to avoid colliding with Shuttle 1.

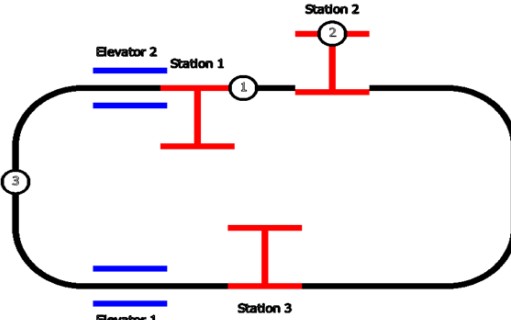

(**c**) Shuttle 2 continues to do the job after Shuttle 1 runs by.

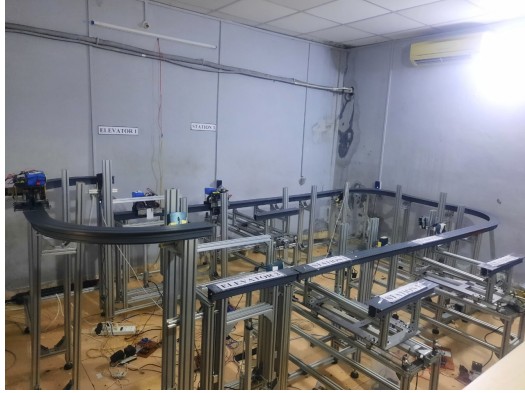

(**d**) The whole real experimental system.

**Figure 25.** Case 4—The low priority shuttle should find a station to avoid a high priority.

After Shuttle 1 runs by Shuttle 2, Shuttle 2 continues to do the job, as shown in Figure 25c.

## 6. Primary Experimental Results

In this section, we will evaluate the control algorithm in the simulation cases in Section 4 by using a real shuttle, station, and elevator. Before entering the test, Figure 25d shows the whole system of stations, lifts, and roads.

### 6.1. Case 1—The Control Algorithm in the Simulation Case with Real Shuttles, Stations, and Elevators

In Figure 26, after the system has been booted for some time, a first task is added (delivery from Station 1 to Station 3). Immediately, the system calculates and selects Shuttle 1 for this new mission (in Figure 27). After selecting, Shuttle 1 begins running to the station to load the goods (Figure 28).

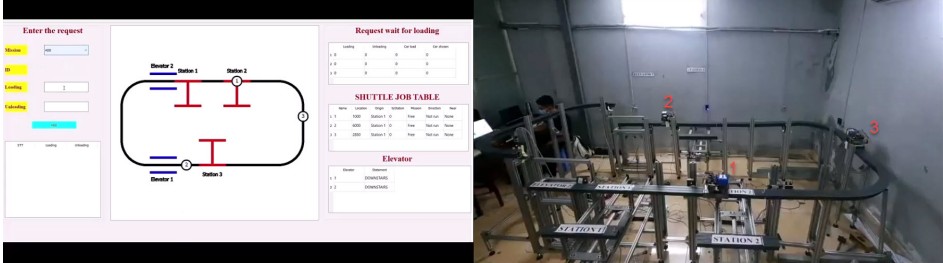

**Figure 26.** The new mission is added from Station 1 to Station 3.

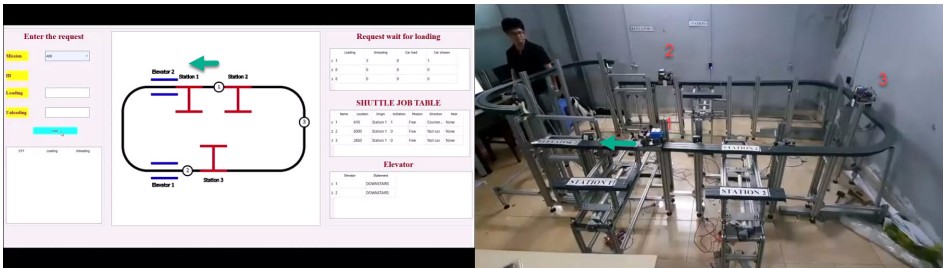

**Figure 27.** The system chooses Shuttle 1 for the mission.

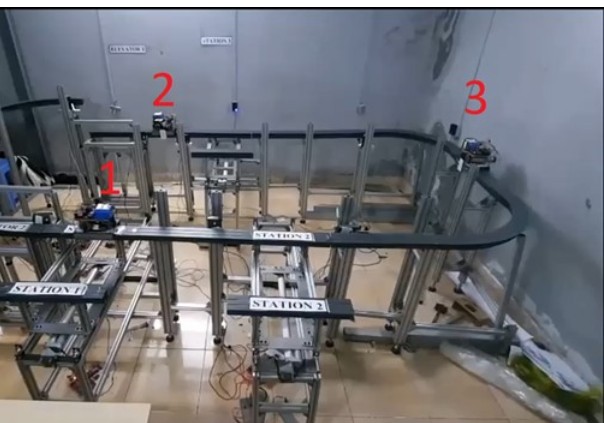

**Figure 28.** The Shuttle 1 is running to Station 1 to get the goods.

In Figure 29, once Shuttle 1 has arrived at Station 1, the electric cylinder of Station 1 starts and brings Shuttle 1 to Station 1 to collect the goods. Shuttle 1 waits to load the goods after successfully entering the station, as depicted in Figure 30. Once the loading is finished, Shuttle 1 will return to the route depicted in Figure 31. The computer map shows that Shuttle 1 has not fully returned to the route, even though it is completely out. The reason for this is that the time assigned to the shuttle in and out of the station is more than the cycle of the electric cylinder because of the fact that when the vehicle enters and exits the

Station, the track areas near the Station will be shaken, so after the Shuttle is in and out, it will take some time for the route to stop shaking.

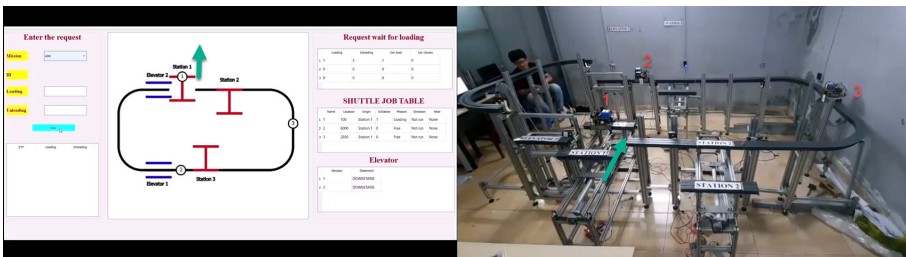

**Figure 29.** Shuttle 1 is going into Station 1 to get the goods.

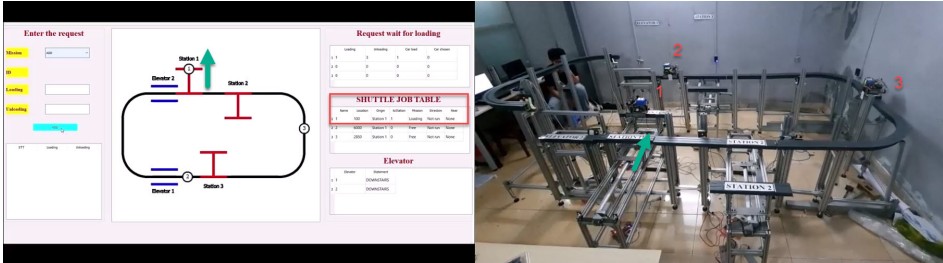

**Figure 30.** The shuttle is loading the goods.

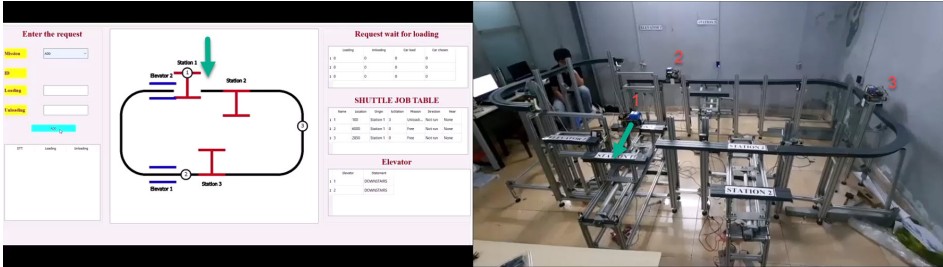

**Figure 31.** The shuttle returns to the route.

In Figure 32, after getting out of Station 1, Shuttle 1 is coming to Elevator 2 (the same as in the simulation). When Shuttle 1 is completely in Elevator 2, the electric cylinder of Elevator 2 starts bringing Shuttle 1 to the upper floor, as shown in Figure 33.

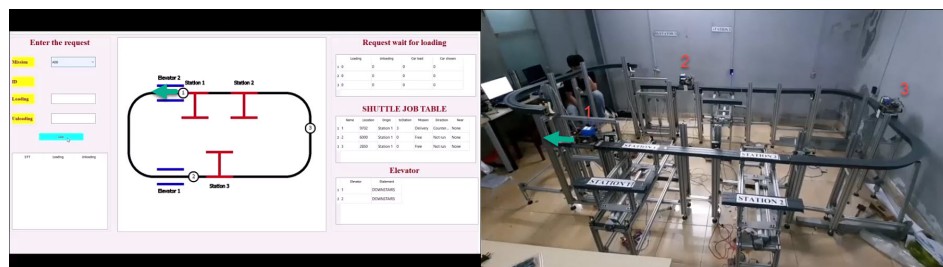

**Figure 32.** Shuttle 1 runs into Elevator 2.

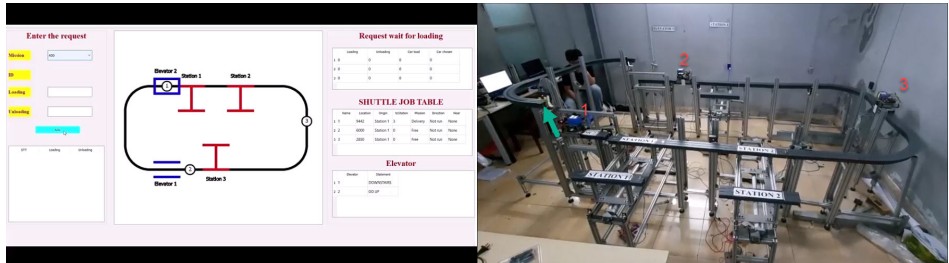

**Figure 33.** Elevator 2 goes upstairs with Shuttle 1.

In Figure 34, Shuttle 1 goes upstairs successfully, and Shuttle 1 keeps going to Station 3 (as in the simulation) by running to Elevator 1 as the Figure 35. However, Elevator 1 is going downstairs, so when Shuttle 1 gets close, it stops and the system starts to activate the electric cylinder to move Ladder 1 up to pick up Shuttle 1, as shown in Figure 36.

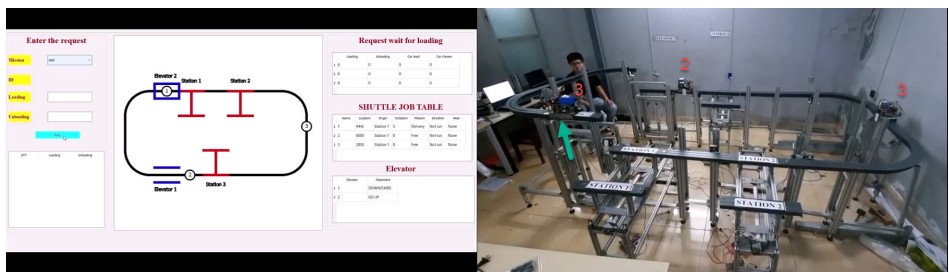

**Figure 34.** Shuttle 1 goes upstairs successfully.

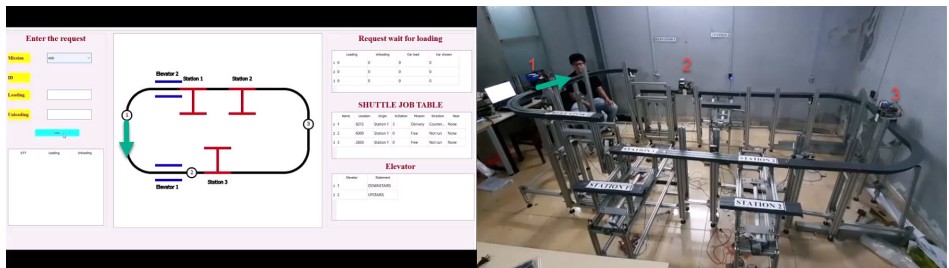

**Figure 35.** Shuttle 1 continues running.

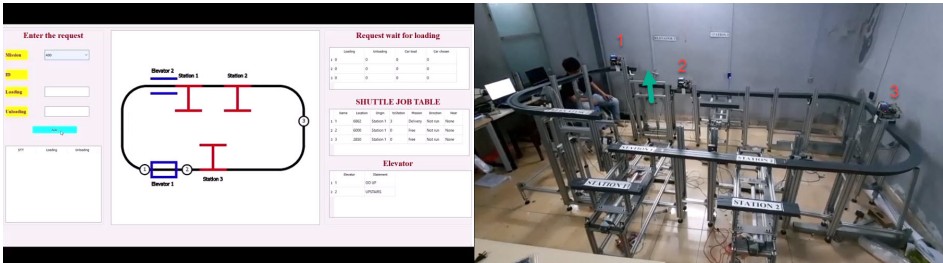

**Figure 36.** Elevator 1 goes up to take Shuttle 1.

After Elevator 1 goes up, Shuttle 1 goes inside Elevator 1, as shown in Figure 37. As depicted in Figure 38, the cylinder activation system went down after that. Finally, Shuttle 1 has completely reached downstairs, as shown in Figure 39.

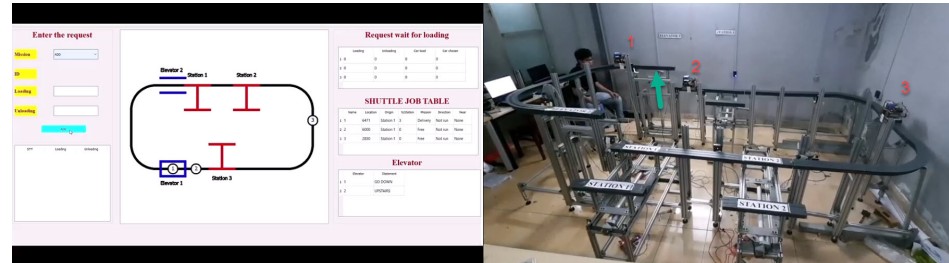

**Figure 37.** Shuttle 1 goes into Elevator 1 to go down.

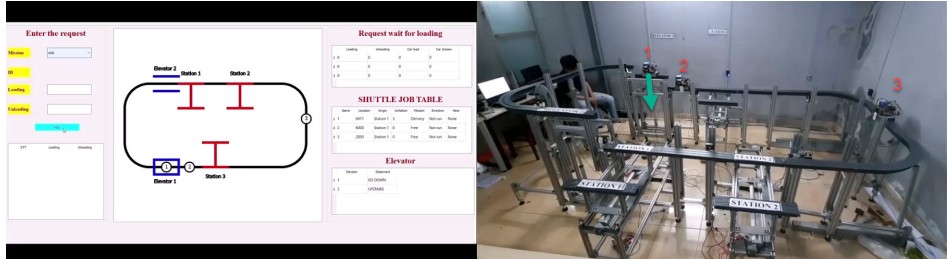

**Figure 38.** Elevator 1 brings Shuttle 1 go down.

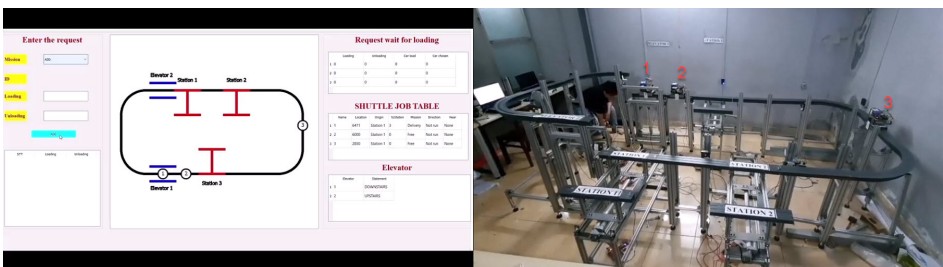

**Figure 39.** Shuttle 1 successfully goes down.

According to Figure 40, at that moment, the distance between Shuttle 1 and Shuttle 2 is shorter than the safe distance between the two shuttles. The system sends a command to capture the group of Shuttle 2 with Shuttle 1, and Shuttle 2 is instructed to run away with Shuttle 1. However, because the transmission signal is a TCP/IP signal, it takes a while for Shuttle 2 to receive the command to start running, so we can see that Shuttle 1 is close to Shuttle 2, as is shown in Figure 41. However, because the safe distance between two shuttles is big enough, no accidents occur.

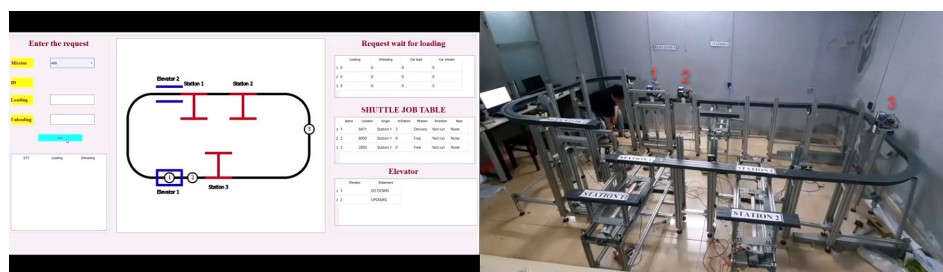

**Figure 40.** System orders Shuttle 2 to run with Shuttle 1.

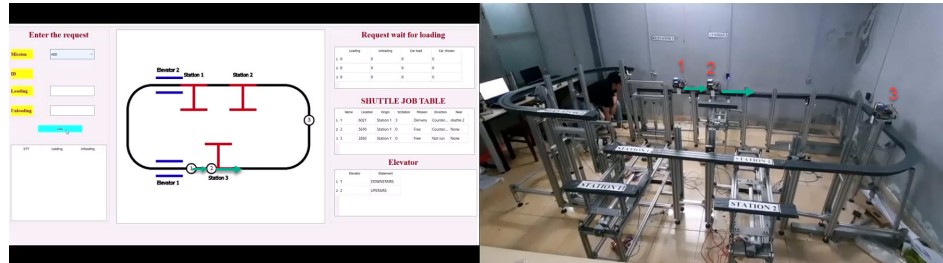

**Figure 41.** Shuttle 2 runs with Shuttle 1.

In Figure 42, Shuttle 1 goes into Station 3 successfully but Shuttle 2 does not go outside of Station 3, so Shuttle 2 keeps running until getting out of Station 3. After Shuttle 2 is outside, the system makes Station 3 bring Shuttle 1, as shown in Figure 43.

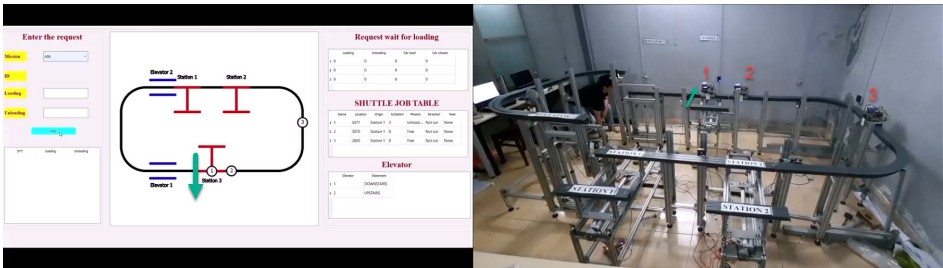

**Figure 42.** Shuttle 2 keeps running.

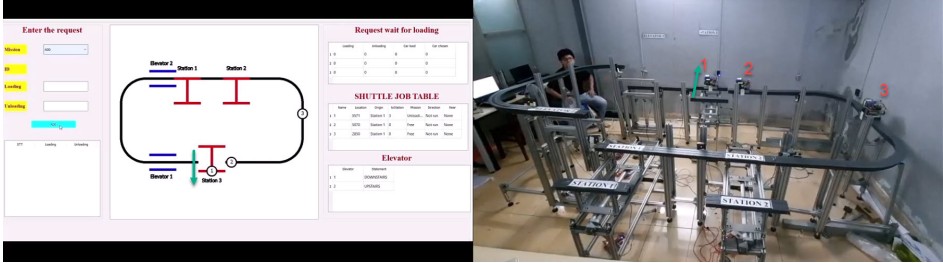

**Figure 43.** Shuttle 1 goes into Station 3.

In Figure 44, the goods in Shuttle 1 are delivered, after time is out, Shuttle 1 returns to the route. To conclude case 1, the system behaves as in the simulated results. However, the time required for getting in and out of the station and going up and down in the elevator does not match the simulation perfectly due to the influence of mechanics and signal transmission.

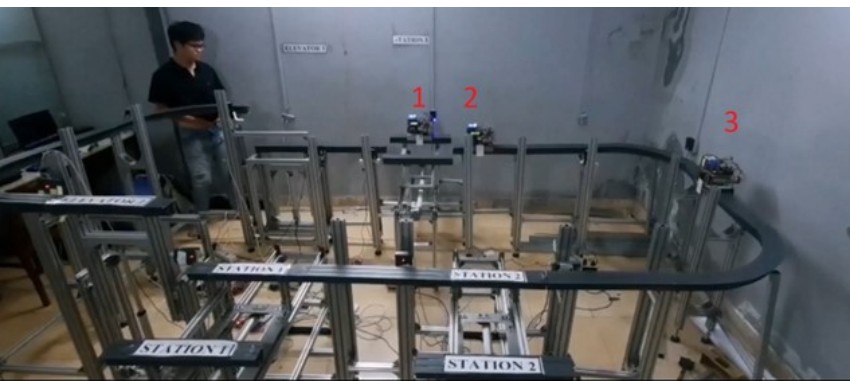

**Figure 44.** After unloading, Shuttle 1 returns to the route.

### 6.2. Case 2—The Shuttle Doing Its Job Must Making Another Shuttle Run Away

In Figure 45, the new mission of delivering goods from Station 2 to Station 3 is added by the users. Immediately, the system calls Shuttle 1 for this mission, which is shown in the "Shuttle chosen" column in the "Request wait for loading" table in the Figure 46, and Shuttle 1 runs to Station 2. Everything is the same as in the simulation in Section 4. After Shuttle 1 arrives at Station 2, the electric cylinder activates and moves Shuttle 1 into the loading position.

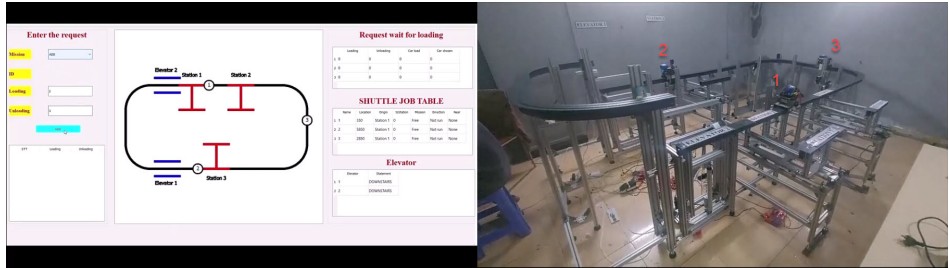

**Figure 45.** The new mission is added.

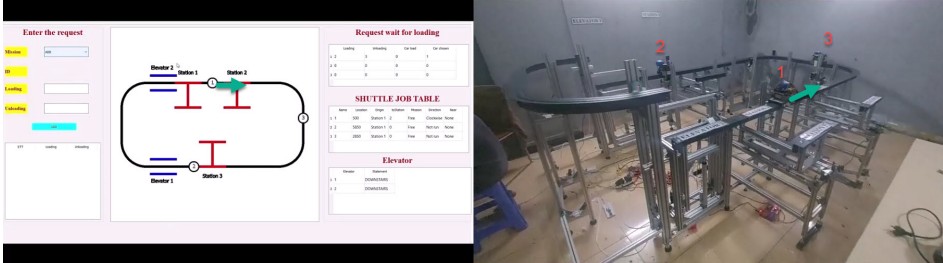

**Figure 46.** Shuttle 1 goes to Station 2 to acquire the goods.

During the Station 2 receives Shuttle 1 in Figure 47, Shuttle 1 gets out of Station 3 after loading the goods as Figure 48. The status of Shuttle 1 switches to needing to unload in the "Shuttle job table". After finishing the exit, the system calculates the direction of Shuttle 1, as shown in the column direction in the table entitled "Shuttle job table" in Figure 49. At the same time as in Figure 49, Shuttle 1 runs.

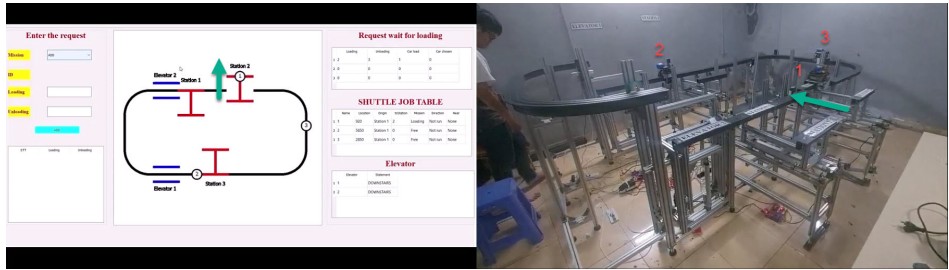

**Figure 47.** Station 2 receives Shuttle 1.

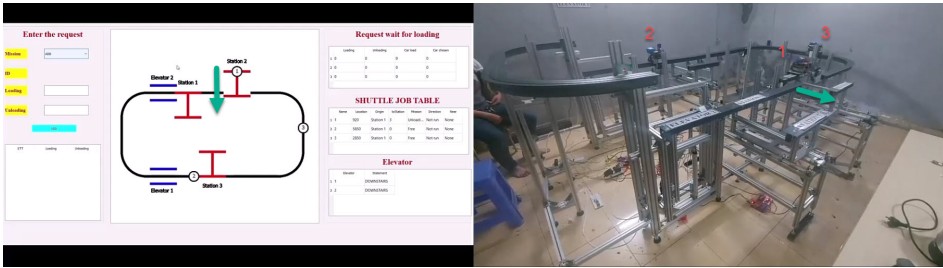

**Figure 48.** Shuttle 1 returns to the route for delivering.

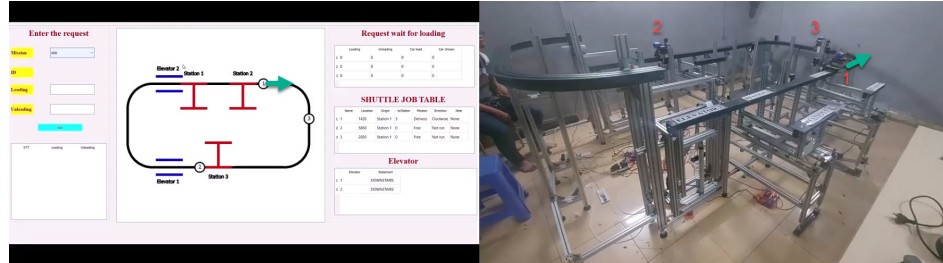

**Figure 49.** Shuttle 1 begins running to Station 3 for delivering.

As in case 1, while Shuttle 1 is running to Station 3, it meets Shuttle 3 and forms a group with Shuttle 3 (Shuttle 3 has lower priority compaered to Shuttle 1). So, Shuttle 3 and Shuttle 1 run to Station 3,as shown in Figure 50. When nearly at Station 3, the distance between Shuttle 2 and Shuttle 3 is shorter than the "safe distance", so Shuttle 2 is added to the group and runs with the others, as is seen in Figure 51.

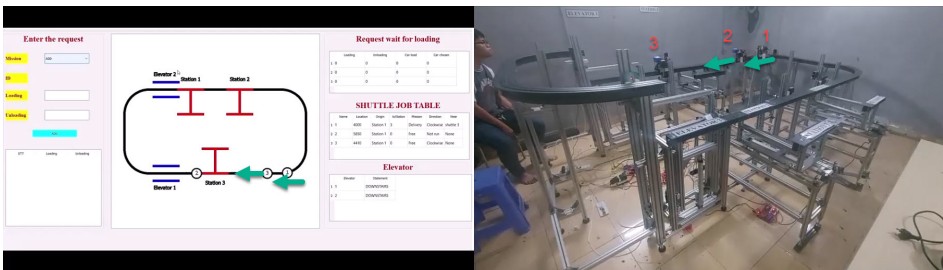

**Figure 50.** Shuttle 3 is forced to run by Shuttle 1.

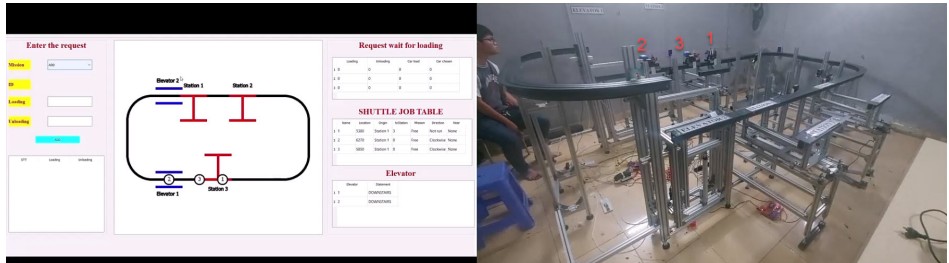

**Figure 51.** Shuttle 2 is forced to run by the Shuttle 1 group.

Figure 52 shows that after Shuttle 1 has arrived at Station 3, the distances of the others from Station 3 are too far. The system enables the electric cylinder in Station 3 to function and transports Shuttle 1 for unloading. After finishing unloading, the Shuttle 1 returns to the route. In conclusion, in the second case, all operations of the system almost align with those of the simulation, but there are still some deviations in their actual locations compared to the PC map. The unstable wifi network is causing interference in the signal transmission. However, everything remains as expected in the simulation.

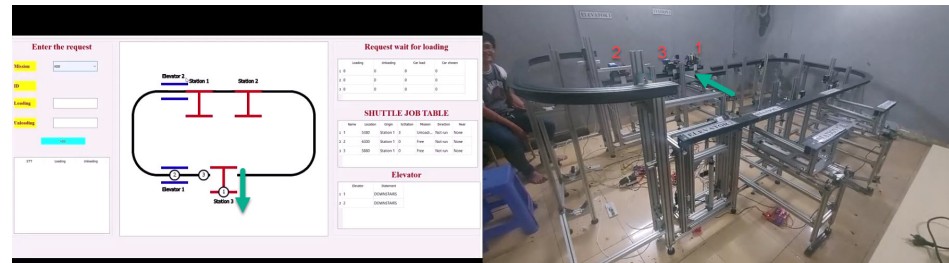

**Figure 52.** Shuttle 1 gets into Station 3.

### 6.3. Case 3—There Is More than One Shuttle Run to Complete the Mission

In this case, there are two shuttles with missions which can interfere with each other during their movement. The shuttle with the lower priority will remain stationary and wait for the shuttle with the higher priority to pass.

In Figure 53, the mission, which is delivered from Station 2 to Station 3, is added by the client. Immediately, the system selects Shuttle 1 for this mission, which can be seen in the "Request waiting for loading" table in column 4 in Figure 54, i.e., the same as in the simulation. After a few seconds, the next mission is added from Station 3 to Station 2, as in Figure 55. At the same time as the mission has appeared, Shuttle 2 is called to load the goods for this new mission (see Figure 56).

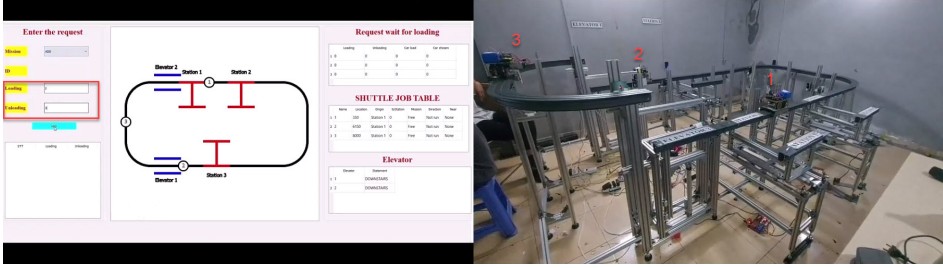

**Figure 53.** The mission from Station 2 to Station 3 is added.

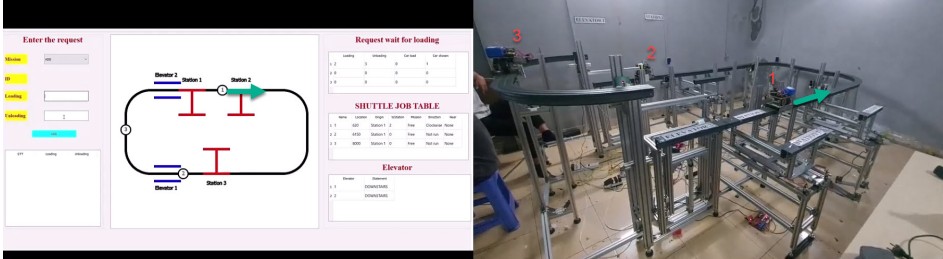

**Figure 54.** Shuttle 1 is chosen for the first mission.

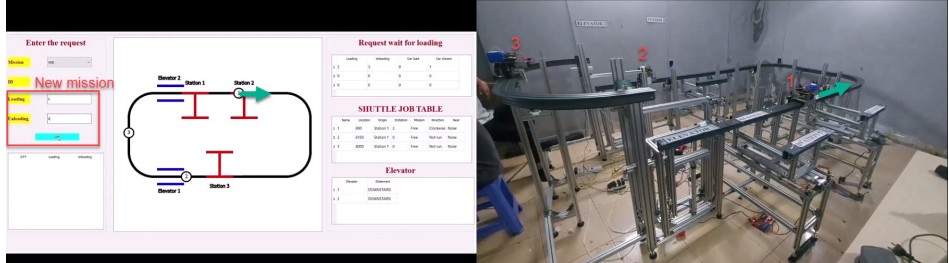

**Figure 55.** The second mission is added.

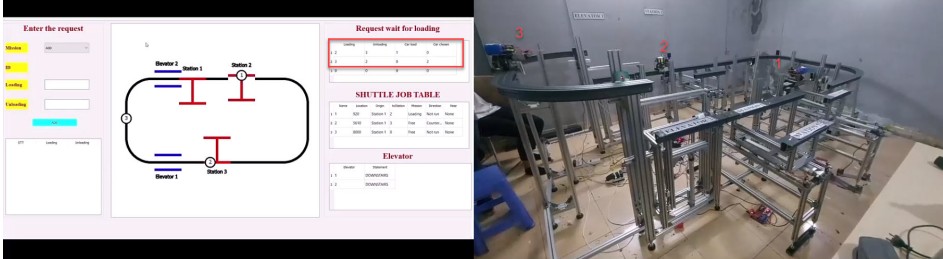

**Figure 56.** The system selects the shuttle for second mission.

In Figure 57, after both Shuttle 1 and Shuttle 2 have arrived at the loading station, the electric cylinders in both stations work to bring both of them for loading. However, Shuttle 1 reaches reached Station 2 before Shuttle 2 reaches Station 3. This is the reason for, as shown in Figure 58, the time of Shuttle 1 returning to the route being earlier than the

time of Shuttle 2. So, while Shuttle 1 begins to deliver, Shuttle 2 is still in Station 3 waiting to exit. We can see the status of the two shuttle in the "Shuttle job table" in Figure 58.

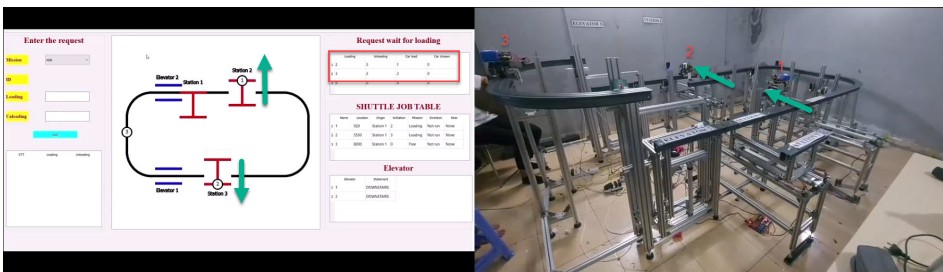

**Figure 57.** Shuttle 1 and Shuttle 2 are loading.

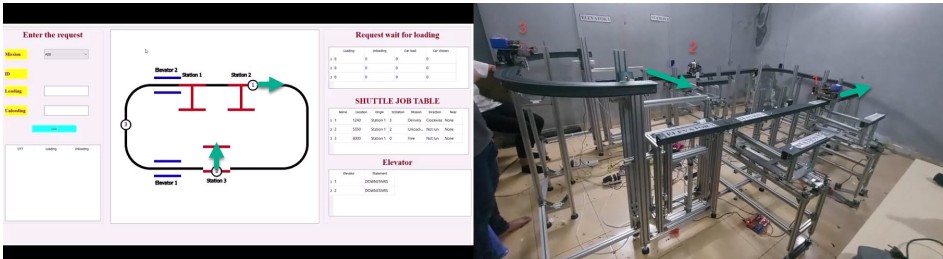

**Figure 58.** Shuttle 1 is coming to Station 3.

In Figure 59, after Shuttle 2 returned to the route, the status of Shuttle 2 changes to "Delivery". However, Shuttle 2 does not run because the best way to unload the food would be to stack it with Shuttle 1. In that way, there are no stations for Shuttle 2 to enter in order to avoid Shuttle 1. This is the reason that the system makes Shuttle 2 stay. When Shuttle 1 wants to go into Station 3, Shuttle 2 must get away from it in order to get into it, as in Figure 60. In Figure 61, after Shuttle 1 has entered Station 3 for unloading, there is no shuttle that can get stuck with Shuttle 2. So, the system makes the decision that Shuttle 2 can run to its unloading station, the same as in the simulation (Section 5). After a few seconds, Shuttle 2 arrives at its unloading station (Station 2) (see Figure 62) and begins unloading.

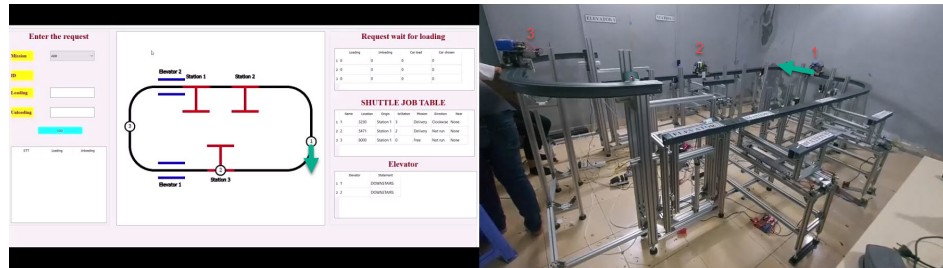

**Figure 59.** The decision of system after Shuttle 2 has returned the route.

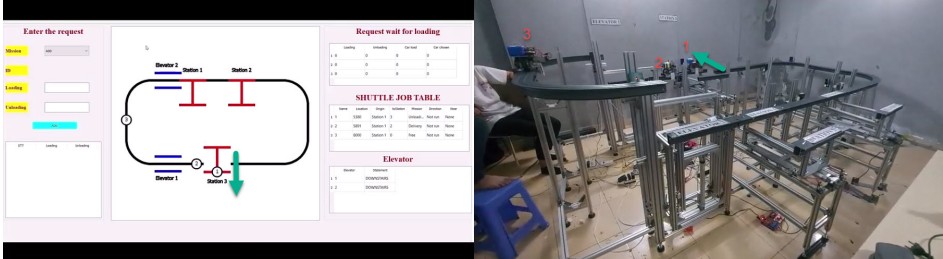

**Figure 60.** Shuttle 2 exists the path of Shuttle 1.

The conclusion of this case is that the simulation is accurate. However, there is still a slight difference, which is the difference between the reality and the picture in the PC, as shown in Figures 58–61.

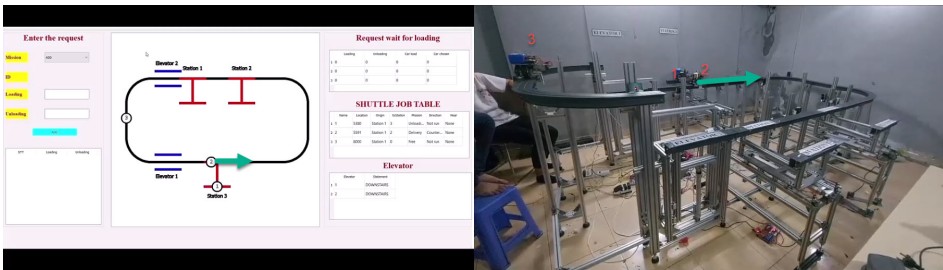

**Figure 61.** Shuttle 2 runs after waiting for Shuttle 1 to enter Station 3.

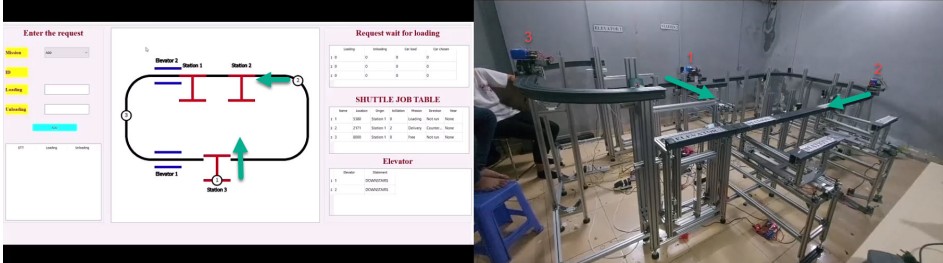

**Figure 62.** Shuttle 2 arrives in the unloading station.

## 7. Discussion

Table 19 provides a comprehensive summary of the research that has been published. Liao, Jeng, and Zhou (2007) [23] and Nakamura et al. (2015) [24] employed a Petri nets (PNs) framework to analyze the behavior of OHT (overhead hoist transport) vehicles within a 300 mm OHT loop. An integer programming methodology was employed by the researchers to effectively capture the occurrence of congestion and the predetermined routes of OHTs. In their study, Huang, Lu, and Fu (2007) [25] employed a genetic algorithm (GA) to identify a blended rule for lot dispatching. In addition, they utilized the Markov decision model to optimize the routing of OHT vehicles. Despite demonstrating dynamic routing algorithms, the researchers used a separate overhead hoist transport (OHT) system and only accounted for a limited number of OHTs. It should be noted that under a segregated overhead hoist transport (OHT) system, the rails are physically partitioned into various loops, with each loop having a restricted number of OHTs. As a result, research on segregated overhead transportation (OHT) systems usually examines a sample size of less than 20 OHTs.

**Table 19.** Summary of studies on dynamic routing of OHT vehicles in semiconductor fabs [26].

| Reference | Track Layout | Methodology | Number of OHT |
|---|---|---|---|
| Liao, Jeng, and Zhou [23] | Segregated | Petri nets and integer programming | 20 |
| Huang, Lu, and Fu [25] | Segregated | Markov decision model | 10 |
| Yang et al. [27] | Unified | K-shortest path algorithm | 20 |
| Bartlett et al. [28] | Dijkstra with dynamic edge weight | 250 programming | 6 |

Yang et al. (2008) [27] introduced a novel approach to dynamic routing in a unified overhead hoist transport (OHT) system within the context of 300 mm wafer fabs. By adapting to the dynamic traffic environment, the dynamic routing method reduces traffic congestion and achieves fast lot delivery. The MOGPRG and dynamic routing method improve mean cycle time and tardy rate, demonstrating their effectiveness in fan performance. In their study, Bartlett et al. (2014) [28] presented a congestion-aware dynamic routing strategy that was demonstrated in automated material handling systems

(AMHS) for semiconductor manufacturing wafer transport. Benzoni et al. (2023) [29] had also developed a look-ahead dispatching algorithm considering arrival times of future requests and vehicle availability. A Mixed integer linear programming (MILP) model and discrete-event simulation model are tested, showing proactive dispatch improves AMHS management by decreasing lot waiting time.By rerouting vehicles efficiently when the congestion status changes, this strategy improves the steady-state routing performance, reduces heavy congestion frequency, and recovers from vehicle breakdowns more efficiently. The algorithm employed in this study involves regularly updating the predicted traversal time of an edge each time a vehicle travels over it. Moreover, a Dijkstra algorithm is used to determine the optimum path at each intersection encountered by OHT cars. The researchers conducted high-fidelity simulation studies on a practical-scale track network and observed a significant decrease in the occurrence of heavy congestion.

## 8. Conclusions

Our research introduces an innovative approach to the overhead Hoist transportation system, a novel conceptual automation system for overhead transportation. The OHT will offer tremendous potential for flexible production processes in reconfigurable factories in the near future. Shuttles are capable of being assigned manufacturing steps and transport tasks. Furthermore, the system has numerous advantageous features that can be utilized in industrial applications, such as logistics and vertical farming. The vehicles can easily and quickly complete their assigned tasks with a simple and effective structure. Furthermore, we were able to create an energy optimization algorithm, which is a positive aspect of this study. In the future, we aim to fully utilize the working space, develop more modules for transferring between rails, and introduce many other functional modules. The results are the starting point for a high-performance follow-up system, involving a larger modular rail grid and multiple mobile units with manipulators mounted to develop increasingly complex applications (involving manipulation tasks, navigation, collision avoidance, and motion planning in a real multi-shuttle transportation system). An increasing number of control algorithms, experiences, and innovative concepts are being implemented to improve the quality of the entire system in the near future.

**Author Contributions:** T.Q.V. proposed the idea to design the mechanical modulue, instructed the constructing of the controller, and proposed the analysis of the feasible cases of operation for the whole system. T.Q.V. corrected the first manuscipt of this issue and also the revision version of this manuscript. X.T.N. and T.D.T. designed, fabricated and assembled the whole practical system and conducted the experiments. They both contributed to compose this manuscript and also contributed to the revision verion of this manuscript. T.A.V. contributed to the survey of other researches relating to this study. T.T.C.V. and H.B.T. designed and fabricated the controller for the whole system. They also played a role in collecting the data when doing the experiments on the whole system. N.H.L.K. and Q.D.L. developed the simulation to check the operation of the whole system on a computer and proposed the control parameters. All authors have read and agreed to the published version of the manuscript.

**Funding:** This research is funded by Ho Chi Minh City University of Technology (HCMUT), VNU-HCM under grant number B2021-20-04.

**Institutional Review Board Statement:** Not applicable.

**Informed Consent Statement:** Not applicable.

**Data Availability Statement:** The whole datasets proposed in this manuscript are not publicly available, but they are available from the corresponding author on reasonable requests.

**Acknowledgments:** This research is funded by Ho Chi Minh City University of Technology (HCMUT), VNU-HCM under grant number B2021-20-04. We acknowledge Ho Chi Minh City University of Technology (HCMUT), VNU-HCM for supporting this study.

**Conflicts of Interest:** The authors declare that that have no conflict of interest.

## Abbreviations

The following abbreviations are used in this manuscript:

| | |
|---|---|
| AGV | Automated guided vehicle |
| AMHS | Automated material handling systems |
| BCG | Boston Consulting Group |
| CMEM | Comprehensive model emission model |
| GA | Genetic algorithm |
| hazmat | Hazardous materials |
| HCMUT | Ho Chi Minh City University of Technology |
| IRP | Inventory routing problem |
| SCLP | Line sensor location problem |
| LIRP | Location inventory routing problem |
| MSMSTS | Multi-station multi-shuttle transportation systems |
| MHND | Multi-class hazmat distribution network design |
| OHTS | Overhead hoist transportation systems |
| O/D | Origin/destination |
| PNs | Petri nets |
| PDF | Probability density function |
| RFID | Radio frequency identification |
| STN | Stochastic transportation network |
| TSLP | Traffic sensor location problem |
| VRP | Vehicle routing problem |
| VNU-HCM | Vietnam National University, Ho Chi Minh City |

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
