# Peer review of "A Study on the Design and Control of the Overhead Hoist Railway-Based Transportation System"

_applsci, doi:10.3390/app13179985_

Round 1

Reviewer 1 Report

This paper proposes an interesting shuttle system with pickups and delivery for logistics systems. There are many results and pictures. The experiments section is long and needs to be clarified. In addition, the literature review is less comprehensive. Some recent studies on logistics are missing, just to name a few,  Multi-class hazmat distribution network design with inventory and superimposed risks. Transportation Research Part E, 2022, 161, 102693. A hybrid metaheuristic algorithm for location inventory routing problem with time windows and fuel consumption. Expert Systems With Applications, 2021, 166, 114034

The paper is generally well written.

Author Response

Dear Applied Sciences Editorial Office and Dear Reviewer,

Subject: Submission of revised manuscript ID applsci-2542107 entitled “A STUDY ON DESIGN AND CONTROL OF THE OVERHEAD HOIST RAILWAY BASED TRANSPORTATION SYSTEM”.

We would like to thank you very much for your email enclosing the reviewer’s comments. After carefully reading the comments, we have revised the manuscript accordingly. Below, we provide our responses in a point-by-point manner. Modifications of the revised manuscript are indicated in Italic.

Our expectation is that the revised version is now appropriate for the requirements of Applied Sciences and we are looking forward to hearing from you soon.

Best Regards,

Assoc.Prof. Tuong Quan Vo, PhD

Director, Bach Khoa Research Center for Manufacturing Engineering.

Ho Chi Minh City University of Technology (HCMUT), Viet Nam National University Ho Chi Minh City – VNUHCM.

On behalf of all of the co-authors.

Reviewer 2 Report

This manuscript uses a constructed model to study different controller schemes of the overhead hoist railway-based transportation system. In my opinion, the article's structure and methodology misses some major points.

·        The authors are advised to place a table to compare different OHTS architectures.

·        The authors should devote a separate section to illustrate the problem's complexity and dimension in a general way.

·        A mathematical formulation with a clear objective function would be necessary for the reader to track what the presented algorithms are doing.

·        The case study should be presented first before starting to illustrate the cases.

·        The deterministic solution scheme is questionable. Is there a possibility to incorporate Artificial Intelligence to optimize the controller like the reinforced learning technique of Genetic Algorithm?

·        No comparison results with other systems to show the superiority of the presented one.

·        Some supported references should be mentioned for the Dijkstra algorithm and Bellman equations like: "Pareto optimal path generation algorithm in stochastic transportation networks." "Application of Dijkstra algorithm in robot path-planning." and "Exact and Heuristics Algorithms for Screen Line Problem in Large Size Networks: Shortest Path-Based Column Generation Approach."

English level is poor and needs major enhancements.

Author Response

(The authors gave the same response as above.)

Reviewer 3 Report

The article deals with the issue of advanced technologies in railway transport management. This is a topical issue and research and development in this area is much needed. The scientific contribution of this paper is the field of applied sciences is indisputable. Recommendations to the authors are to restructure the paper into a standard scientific paper and to add references.

Author Response

(The authors gave the same response as above.)

Round 2

Reviewer 1 Report

This is fine

Reviewer 2 Report

No further comments are required.

None.